# Molecular mechanism of parental H3/H4 recycling at a replication fork

Fritz Nagae[1], Yasuto Murayama ®[2,3] & Tsuyoshi Terakawa ®[1]✉

In chromatin replication, faithful recycling of histones from parental DNA to replicated strands is essential for maintaining epigenetic information across generations. A previous experiment has revealed that disrupting interactions between the N-terminal tail of Mcm2, a subunit in DNA replication machinery, and a histone H3/H4 tetramer perturb the recycling. However, the molecular pathways and the factors that regulate the ratio recycled to each strand and the destination location are yet to be revealed. Here, we performed molecular dynamics simulations of yeast DNA replication machinery, an H3/H4 tetramer, and replicated DNA strands. The simulations demonstrated that histones are recycled via Cdc45-mediated and unmediated pathways without histone chaperones, as our in vitro biochemical assays supported. Also, RPA binding regulated the ratio recycled to each strand, whereas DNA bending by Pol ε modulated the destination location. Together, the simulations provided testable hypotheses, which are vital for elucidating the molecular mechanisms of histone recycling.

Eukaryotic DNA forms chromatin, a linear array of nucleosomes, each composed of one H3/H4 tetramer and two H2A/H2B dimers wrapped by 147 base-pairs (bp) DNA[1]. These nucleosomes modulate DNA transactions such as DNA replication, transcription, and repair by permitting or occluding access of regulatory proteins to DNA depending on histone variants or histone post-translational modifications[2,3]. Therefore, nucleosomes and their constituent histones carry epigenetic information that regulates DNA transactions[3–5]. In chromatin replication, DNA replication machinery (replisome) composed of Mcm2-7, Cdc45, GINS, Pol ε, and RPA inevitably collides with a nucleosome, which must be dismantled to allow the helicase to pass through the nucleosome array[6–8]. Subsequently, the dismantled histones are recycled to replicated leading or lagging strands[6–10]. Faithful histone recycling is vital for maintaining epigenetic information across generations[4,5,11–13].

A recent experimental study revealed that disrupting the interaction between the N-terminal intrinsically disordered tail (N-tail) of Mcm2, a subunit of Mcm2-7, and a histone H3/H4 tetramer perturbs faithful histone recycling[14]. Indeed, the Mcm2 N-tail interacted with the H3/H4 tetramer in the crystal structure[15,16]. Also, previous in vitro biochemical assays showed that the Mcm2 N-tail promotes

nucleosome assembly on DNA[17], suggesting that the Mcm2 N-tail directly deposits the H3/H4 tetramer to the replicated strands. As a support, these interactions are essential for preserving heterochromatin silencing at sub-telomeric loci[18]. Recent experimental studies also demonstrated that nucleosomes were assembled on the replicated strands in the presence of histone chaperones and chromatin remodelers[19,20]. However, it remains unknown whether the Mcm2 N-tail alone is sufficient to hand over the H3/H4 tetramer to the replicated strands or whether additional histone chaperones are necessary. Also, the molecular pathways via which the H3/H4 tetramer attached to Mcm2 is recycled to the replicated strands have yet to be deciphered.

Symmetric histone recycling to the two replicated strands underlies the maintenance of cellular identity after cell division[4,14,18,21–23]. In contrast, asymmetric histone recycling alters gene expression profiles in the two daughter cells and potentially triggers cell differentiation[4,5,24]. Therefore, the mechanism to regulate the ratio recycled to each strand is vital for organisms to maintain or change their cellular state. Also, the destination location may fine-tune the gene expression or necessitate dramatic nucleosome remodeling after chromatin replication. However, the factors that regulate the

[1]Department of Biophysics, Graduate School of Science, Kyoto University, Kyoto, Japan. [2]Department of Chromosome Science, National Institute of Genetics, Shizuoka, Japan. [3]Department of Genetics, Graduate University for Advanced Studies (SOKENDAI), Shizuoka, Japan. ✉e-mail: terakawa@biophys.kyoto-u.ac.jp

ratio recycled to each strand and the destination location still need to be discovered.

Hitherto, deep sequencing of chromatin in nucleus revealed the ratio recycled to each strand and the nucleosome position after chromatin replication in vivo[14,21,22,25,26]. However, since various biomolecules are mixed in a nucleus, it has been difficult to clarify the molecular mechanism by which minimal factors achieve histone recycling. An in vitro experiment was also performed to reconstitute the recycling reaction with only purified components[19]. However, the study did not analyze the ratio and the destination location. Previous cryo-electron microscopy studies determined the structures of the DNA replication machinery engaging the replicated DNA[27–30]. However, these static structures lack flexible regions, including the Mcm2 N-tail and the replicated leading and lagging DNA strands. Also, it has been challenging to elucidate the dynamics of the H3/H4 recycling. Overcoming these challenges requires visualization of the molecular structural trajectory from H3/H4 bound to Mcm2 until handed over to the replicated strands.

Coarse-grained molecular dynamics simulations have successfully visualized the dynamics of protein-DNA complexes in long-time scale, providing insights into molecular mechanisms[31–35]. In this study, we first performed coarse-grained molecular dynamics simulations of yeast DNA replication machinery containing Mcm2-7, Cdc45, GINS, Pol ε, and RPA, bound to an H3/H4 tetramer, and replicated DNA strands. Previous studies carefully calibrated the coarse-grained model to reproduce the intrinsically disordered tail dynamics[36], electrostatic interactions[37], and nucleosome assembly dynamics[38,39], all required for the current histone recycling simulations. In this study, we also calibrated the interaction parameters of the Mcm2 N-tail and the H3/H4 tetramer. The simulations demonstrated that H3/H4 tetramers can be recycled to replicated strands without histone chaperones, as supported by in vitro replication assays using purified proteins in the current study. The simulation trajectories also revealed two dominant pathways for histone recycling: the Cdc45-mediated and unmediated pathways. In the Cdc45-mediated pathway, the H3/H4 tetramer is once bound to Cdc45 and handed over to the leading strand. On the other hand, in the Cdc45-unmediated pathway, the tetramer is directly handed over to the lagging strand without binding to Cdc45. Consistent with the Cdc45-mediated pathway, the native-polyacrylamide gel electrophoresis (Native-PAGE) assays confirmed that Cdc45 and H3/H4 tetramer electrostatically interact with each other. Also, RPA binding to the ssDNA portion of the lagging strand regulated the ratio recycled to each strand, whereas DNA bending by Pol ε modulated the destination location. Together, the simulations provided valuable insights and experimentally verifiable hypotheses concerning the molecular mechanism in vitro, which is crucial for elucidating the mechanism of in vivo histone recycling regulated by the collaborative actions of multiple histone chaperones.

## Results

### Modeling of the replicated-DNA-engaged replisome binding to the H3/H4 tetramer

This study adopted the AICG2+ model for proteins (see the original paper[40] for details) in which one particle at the $C_\alpha$ atom represents one amino acid and the 3SPN.2 model for DNA (see the original paper[41] for model details). in which one nucleotide is represented by three particles placed at the base, sugar, and phosphate sites. Potential energy functions for protein-protein and protein-DNA interactions model excluded volume and electrostatics. These potential energy functions have successfully reproduced the dynamics of the bacterial architectural protein HU[42], the histone chaperone Nap1[43], and the DNA mismatch recognition protein MutS along DNA[44] in previous studies. To model the interaction between the Mcm2 N-tail and H3/H4, we added a function (a so-called Gō-like potential) stabilizing the reference native protein structure[15]. The parameters of this potential play a

decisive role in H3/H4 competition between Mcm2 N-tail and replicated strands in histone recycling. Therefore, in this study, we performed temperature replica exchange simulations of Mcm2 and an H3/H4 dimer associating to and dissociating from each other with varying parameters (Supplementary Fig. 1) and selected the one that reproduced the experimental binding free energy (Simulation: $-10.00 \pm 0.26$ kcal/mol, Experiment: $-10.45 \pm 0.04$ kcal/mol[15]). To model the interaction between the H3/H4 tetramer and DNA, we added a hydrogen bonding potential as in the previous studies that successfully recapitulated nucleosome stability and DNA unwrapping dynamics[38,39].

We used the crystal and the cryo-EM structures as a reference to prepare the initial structure of each subunit (listed in Fig. 1A) of a replisome binding to the H3/H4 tetramer (see "Methods" for detail). We modeled the replicated DNA strands based on the DNA conformation in the cryo-EM structure (PDB ID: 6U0M[27]). The leading and lagging dsDNA region was extended to 90 and 96 bp, sufficient to wrap around the H3/H4 tetramer to form a tetrasome by superimposing the ideal B-form dsDNA structures (Fig. 1B). To assemble all the components, we manually placed the H3/H4 tetramers and the two RPA molecules proximal to the Mcm2 N-tail and the 59 nt lagging ssDNA region[45], respectively, and performed the equilibration molecular dynamics simulation for $1 \times 10^6$ steps so that these molecules associate with their binding site. Notably, previous single-molecule fluorescence imaging[46] showed the two RPA molecules at the replication fork in the physiological concentration of DNA polymerase α (Pol α)[47]. The Mcm2 N-tail bound to the H3/H4 tetramer (residues 1–200) as in the crystal structure (PDB ID: 4UUZ[15]) and occupied the DNA binding interface of the H3/H4 tetramer (Fig. 1C). The recent cryo-EM structure of an endogenous replisome[48] suggested the relevance of interactions between the H3/H4 tetramer and the Mcm2 N-tail as intermediate structures in histone recycling reactions within cells. Here, we focused on the recycling pathways in which the H3/H4 tetramer binds to the Mcm2 N-tail, although other pathways may also be possible. Starting from this initial structure (Fig. 1D), we performed Langevin dynamics simulations. The parameters are the same as those of the previous study in which nucleosome stability and DNA unwrapping dynamics in a physiologically relevant condition were successfully recapitulated[38,39].

### Replisome directly recycles an H3/H4 tetramer to replicated DNA strands

We performed 100 runs of the simulation of a replicated-DNA-engaged replisome binding to an H3/H4 tetramer for $1 \times 10^8$ steps. In 20% (20/100) of the simulation trajectories, the Mcm2 N-tail directly deposited the H3/H4 tetramer onto either of the two replicated strands (Fig. 2A, B, Supplementary Movies 1–4). The other simulation trajectories showed the association of the H3/H4 tetramer with the parental strand (26/100, Supplementary Fig. 2A) or no association (54/100). We estimated that it would take $>1 \times 10^9$ steps for the H3/H4 tetramer to associate with the parental, leading, or lagging strand in all the trajectories (Supplementary Fig. 2B). The H3/H4 tetramer did not dissociate from DNA during the simulations once it was deposited (Fig. 2C, Supplementary Fig. 2C–E), indicating that the tetramer deposited on the parental strand (26/100, Supplementary Fig. 2A, C) may require additional factors such as histone chaperones to be evicted. Below, we focused on the 20 recycling trajectories to statistically analyze the molecular pathway of histone recycling.

Next, we sought to experimentally confirm whether a replisome can recycle histones to replicated DNA strands upon collision with a nucleosome without histone chaperones. For this purpose, we purified the budding yeast histones and the replication-related proteins to biochemically reconstitute DNA replication[49–51] with nucleosome-assembled DNA. We prepared the 1155 bp linear DNA substrate containing Autonomously Replicating Sequence 1 (ARS1) and the Widom

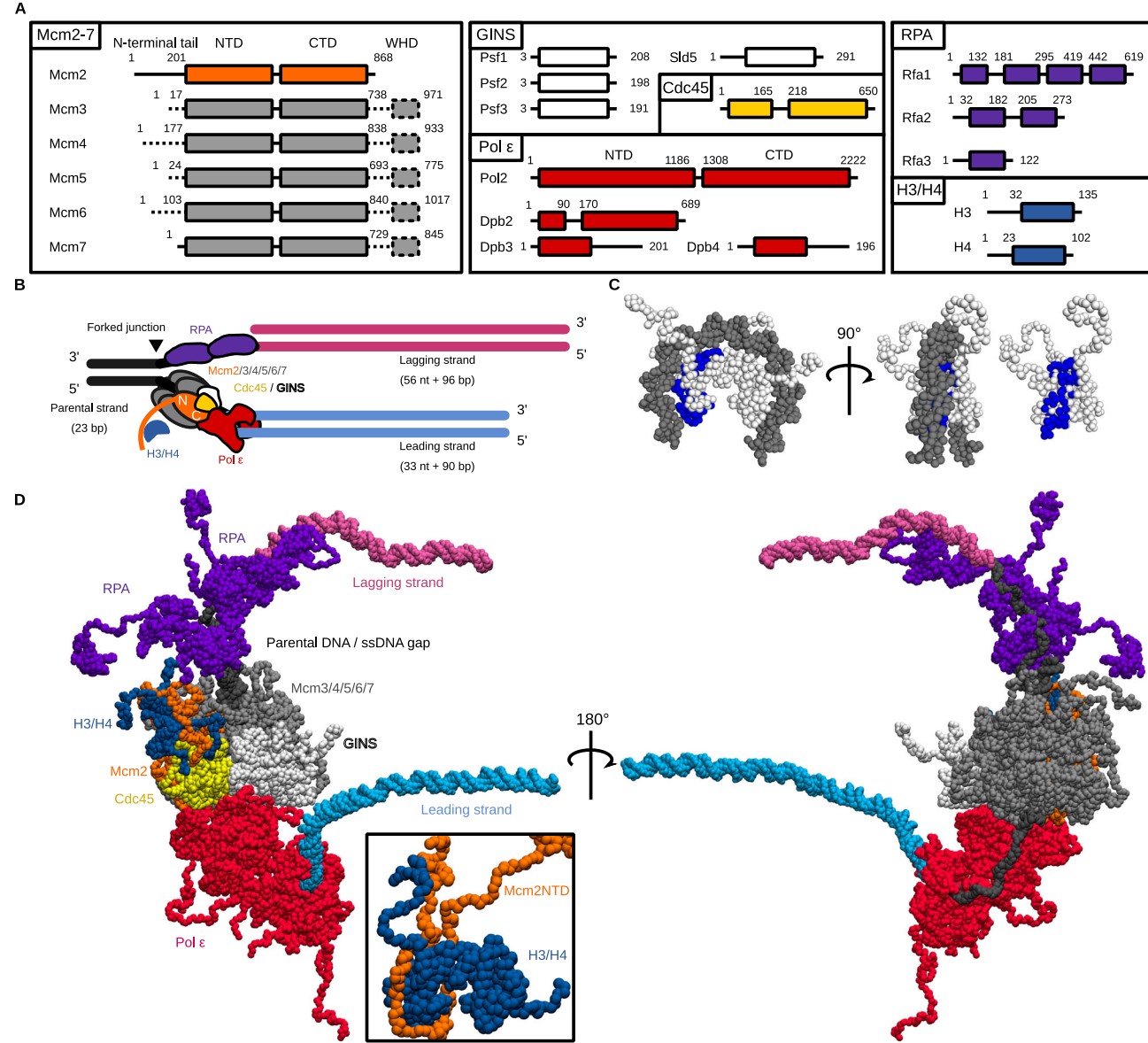

**Fig. 1 | The initial structure for the coarse-grained molecular dynamics simulations of a replicated-DNA-engaged replisome binding to an H3/H4 tetramer.**
**A** The domain composition of Mcm2-7, GINS, Cdc45, Pol ε, RPA, and the H3/H4 tetramer. The dotted lines represent the regions not included in the simulations.
**B** Schematic illustration of the initial structures. The black, pink, and cyan lines represent the parental, lagging, and leading strands. The orange, gray, white, yellow, red, blue, and purple objects represent Mcm2, Mcm3/4/5/6/7, GINS, Cdc45, Pol ε, H3/H4, and RPA, respectively. **C** The structures of a tetrasome and an H3/H4 tetramer. The residues contacting Mcm2 in the crystal structure (PDB ID: 4UUZ) are colored blue. **D** The initial coarse-grained structures of the replicated-DNA-engaged replisome.

601 sequence (Fig. 2D, Supplementary Table 1). The nucleosome was initially reconstituted on the DNA substrate by the salt gradient dialysis method[52], and then the CMG helicases were assembled on the substrate to initiate the replisome-dependent DNA replication. Nascent DNA syntheses were labeled by incorporating biotinylated deoxy uridine nucleotide (biotin-dUTP). After incubation for sufficient time (20 min) for replication to complete (Fig. 2E left, lanes 3 and 7), we treated the reactions with micrococcal nuclease (MNase). This digests nucleosome-free DNA regions, whereas the nucleosome-coated DNA segments were protected from the digestion, generating ~150 bp DNA fragments. The products were analyzed by polyacrylamide gel electrophoresis. We observed ~150 bp biotinylated DNA fragments in the reaction performed with the nucleosome substrate, whereas no detectable signal was seen when naked DNA was used as a substate (compare lanes 4 and 8 in Fig. 2E left). This observation suggested that

nucleosomes were formed (recycled) on the replicated DNA strands. However, this ~150 bp band could be generated by end-labeling of the MNase-digested, non-replicated nucleosomal DNA by Pol ε rather than nucleosome reassembly during the replisome-dependent DNA replication. To rule out the possibility, we repeated the assay without Mcm10, which is essential for DNA replication initiation by replisomes. In this case, no detectable biotinylated ~150 bp band was seen (compare lanes 6 and 8 in Fig. 2E left). On this gel (Fig. 2E right) on which both pre-replicated and post-replicated DNA can be detected, we also observed a similar level of nucleosome assemblies both in the presence and absence of Mcm10, demonstrating that the biotinylated ~150 bp band was not majorly produced by end-labeling. Together, these biochemical assays supported the simulation prediction that a replisome can recycle histones to replicated DNA strands upon collision with a nucleosome without histone chaperones.

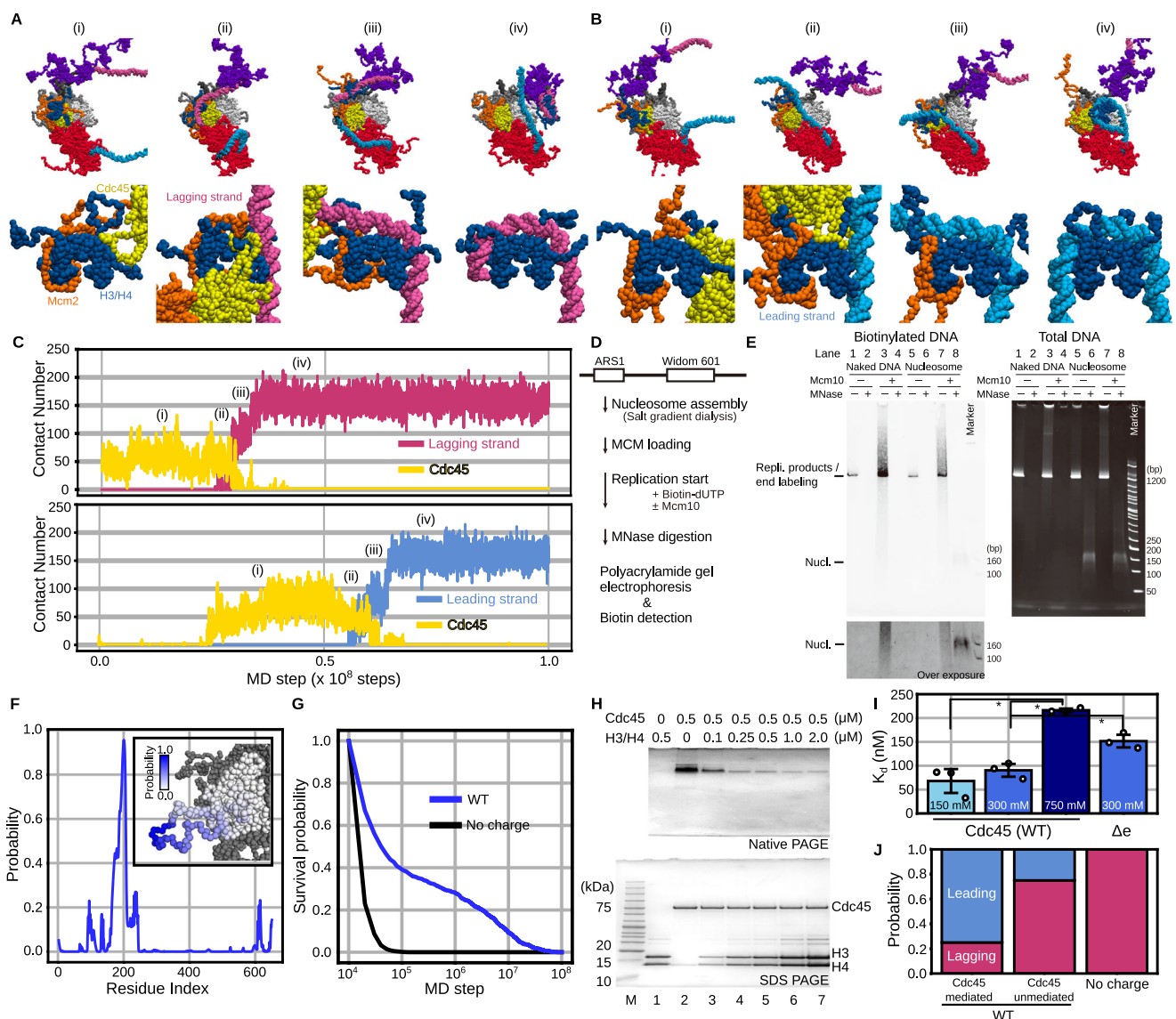

**Fig. 2 | The H3/H4 tetramer was recycled to the replicated strands by the Mcm2 N-tail in the simulations of a replicated-DNA-engaged replisome.** Source data are provided as a Source Data file. **A**, **B** Representative snapshots of the simulation trajectories in which the H3/H4 tetramer was deposited on the lagging (**A**) and leading (**B**) strands. The top panels are the magnified version of the bottom ones. **C** Time trajectories of the number of residues in the H3/H4 tetramer contacting the lagging strand (pink in the top panel), the leading strand (cyan in the bottom panel), and Cdc45 (yellow) from the trajectories in (**A**) and (**B**). **D** Schematic of the in vitro DNA replication with nucleosome-assembled DNA. **E** Gel images of biotinylated DNA (left) and total DNA (right) before and after MNase digestion on native gels. We obtained the similar results from three independent replicates. **F** Probability of each residue in Cdc45 contacting the H3/H4 tetramer in the simulations. The inset

figure is the probabilities represented by shades of blue on the Cdc45 structure. **G** Survival probabilities of the association between Cdc45 and the H3/H4 tetramer in the simulations in the presence (WT) or absence (No charge) of charges in the Cdc45 acidic loop. **H** Gel images of native PAGE (top) and SDS-PAGE (bottom) of the mixtures of Cdc45 and H3/H4 tetramer (lanes 1–7). 'M' denotes a marker (ATTO; 2332346). **I** The apparent dissociation constants between the H3/H4 tetramer and Cdc45 (WT in 150, 300, and 750 mM KCl and Δe (Cdc45Δe) in 300 mM KCl). The error bars represent the mean ± standard deviation from three replicates of each condition. The asterisks represent $P < 0.05$ by two-tailed Welch's t-test (p-values between 150 mM and 750 mM, between 300 mM and 750 mM, and between WT and Δe are 0.013, 0.004, and 0.010, respectively). **J** Ratios of the replicated strands to which the H3/H4 tetramer was recycled.

## An H3/H4 tetramer is recycled via two pathways

Interestingly, in all the trajectories (100/100), we found that the H3/H4 tetramer bound to the Mcm2 N-tail associated with Cdc45 at least once before being deposited to replicated DNA (Fig. 2C, Supplementary Fig. 2D, E). The repeated dissociation from and association with Cdc45 suggested that the simulation result is robust to the initial structure. From the structural point of view, half of the DNA binding surface of the H3/H4 tetramer was wrapped by the Mcm2 N-tail, and the other half was exposed to solvent (Fig. 1C, D). In the simulation trajectories, this exposed surface was associated with Cdc45 [(i) in Fig. 2A–C]. In the recycling trajectories (20/100), either of the leading

or lagging strand fluctuated around the Cdc45-associated H3/H4 tetramer, competed for the binding surface on the H3/H4 tetramer with Cdc45 [(ii) in Fig. 2A–C], and took it away [(iii) in Fig. 2A–C], completing recycling. We calculated the probabilities that each residue of Cdc45 contacts the H3/H4 tetramer and found that residues 184–205 of Cdc45 frequently contacted the H3/H4 tetramer (Fig. 2F). These Cdc45 residues are in the flexible acidic loop not resolved in the cryo-EM structures and associated with the H3/H4 tetramer for ~9 × 10⁶ steps on average (Fig. 2G). As expected from the hypothesis that the interaction between the Cdc45 acidic loop and H3/H4 tetramer is mainly electrostatic interactions, we could hardly observe contacts lasting longer

than $1 \times 10^5$ steps between Cdc45 and the H3/H4 tetramer when the simulations were repeated without the charge in the Cdc45 acidic loop (Fig. 2G). After the binding, in 80% (16/20) of the recycling trajectories, the leading or lagging strand took the H3/H4 tetramer away from Cdc45 (Supplementary Movies 1, 2). In the others (4/20), the Mcm2 N-tail directly deposited the H3/H4 tetramer free from Cdc45 on the leading or lagging strands (Supplementary Movies 3, 4, Supplementary Fig. 2D–G). The rare occurrence of the Cdc45-unmediated pathway suggested that the path is physically feasible but is statistically rare. Together, the simulations showed that the H3/H4 tetramer can be recycled to the replicated strand via the Cdc45-mediated and unmediated pathways.

Next, we sought to perform Native-PAGE assays to experimentally confirm that Cdc45 associates with an H3/H4 tetramer. Thus, we reconstituted H3/H4 tetramers with recombinant histones and purified the budding yeast Cdc45 from *E.coli* as described previously[53] (Supplementary Fig. 3A). We mixed the purified Cdc45 molecules and the reconstituted H3/H4 tetramers, incubated the mixtures for 15 min at 30 °C, and ran the reaction products on native and denaturing gels. In this assay, the H3/H4 tetramers did not enter the native gels due to their high positive net charge (Fig. 2H, lane 1). Interestingly, the intensity of the Cdc45 band (Fig. 2H, lane 2) gradually decreased as the concentration of the H3/H4 tetramers increased (Fig. 2H, lanes 3–7). This is thought to be because Cdc45 no longer enters the gel when it forms a complex with an H3/H4 tetramer due to the high positive net charge of the complex (+40e). Therefore, this result suggested that Cdc45 associates with an H3/H4 tetramer, which is consistent with the simulations.

To examine the contribution of electrostatic interactions to the binding, we repeated the assay with varying KCl concentrations (Supplementary Fig. 3B). Then, we measured the apparent dissociation constants, which were $68 \pm 25$ nM, $90 \pm 14$ nM, and $216 \pm 4$ nM in 150 mM, 300 mM, and 750 mM KCl concentration, respectively (Fig. 2I). The stable association, even in 750 mM KCl, suggested that hydrophobic interactions, which were not modeled in our simulations, also contribute to the complex formation. However, as evident from Supplementary Fig. 4, the surface of Cdc45 is predominantly filled with hydrophilic residues, supporting that the contribution of hydrophobic interaction is not dominant, if any, and justifying not incorporating the interactions into our coarse-grained model. Nevertheless, the experimental result suggested that electrostatic interactions significantly contribute the interaction between Cdc45 and an H3/H4 tetramer in concordance with the simulations.

To more specifically confirm that the acidic loop in Cdc45 contributes to the binding to an H3/H4 tetramer, we repeated the assay with a mutant (Δe) in which all the aspartic and glutamic residues were replaced with the asparagine and glutamine residues, respectively (Supplementary Fig. 3C). Then, we measured the apparent dissociation constants, which were $90 \pm 14$ nM and $152 \pm 13$ nM for the wild-type and mutant proteins, respectively (Fig. 2I). This result suggests that electrostatic interactions between the Cdc45 acidic loop and the H3/H4 tetramer contribute to the complex formation in accordance with the simulations.

Next, we sought to analyze the ratio recycled to each strand in the Cdc45-mediated and -unmediated pathways. Thus, we classified the trajectories based on the recycling pathway and the strand to which the H3/H4 tetramer was deposited. In the trajectories with the Cdc45-mediated pathway, 75% (12/16) and 25% (4/16) of them showed deposition to the leading and lagging strands, respectively (Fig. 2J, Supplementary Movies 1, 2). On the other hand, in the trajectories with the Cdc45-unmediated pathway, 25% (1/4) and 75% (3/4) of them showed deposition to the leading and lagging strands, respectively (Fig. 2J, Supplementary Movies 3, 4). To confirm the effect of Cdc45 binding to the H3/H4 tetramer on the strand bias, we eliminated electrostatic interactions between Cdc45 and the H3/H4 tetramer by

neutralizing the charge of the Cdc45 acidic loop and performed 50 runs of simulations. Of them, only two trajectories (2/50) showed successful histone recycling, both resulting in deposition to the lagging strand (Fig. 2J). These results support that the association between Cdc45 and the H3/H4 tetramer promotes histone recycling, especially to the leading strand.

In histone recycling, the leading or lagging strand takes the H3/H4 tetramer away from the Mcm2 N-tail. Thus, the extent to which the leading and lagging strands probe the H3/H4 tetramer bound to the Mcm2 N-tail by their conformational fluctuations is critical for successful recycling.

To investigate this range, we defined a vector from the center of mass (COM) of Mcm2-7 to the end of the leading or lagging strand. The Z-axis is aligned with the direction of the rotation axis of Mcm2-7, and the X-axis is perpendicular to the Z-axis, passing through the COM of Mcm2 (Fig. 3A). The Y-axis is perpendicular to both X- and Z-axes. The angle between the vectors and the X-axis on the X-Z plane is defined as the elevation angle φ, and the angle between the vectors and the X-axis on the X-Y plane is defined as the azimuthal angle θ (Fig. 3B, C). We assumed that the dsDNA bending only negligibly affects the analysis since the strand length (-90 bp) is shorter than the persistence length (-150 bp)[54]. The analysis showed that the lagging strand tended to orient to the direction with an elevation angle of $4° \pm 36°$ and an azimuthal angle of $160° \pm 79°$ (Fig. 3D). The steric hindrance between the CMG helicase complex and the lagging strand explains the slightly positive mean elevation angle (Fig. 3E). Also, the binding of the lagging strand to the CMG helicase explains why the strand was oriented away from Mcm2. Indeed, the lagging strand was associated with the Mcm3, Mcm5, and Mcm7 zinc finger domains in the simulations (Supplementary Fig. 5) in line with the cryo-EM structure[27]. On the other hand, the leading strand tended to orient to the direction with an elevation angle of $35° \pm 23°$ and an azimuthal angle of $67° \pm 52°$ (Fig. 3F). The association between the leading strand and the N-terminal domain of Pol2, which bent DNA and fixed its orientation, explains this tendency (Figs. 1D, 3G).

Next, we performed a similar analysis using the vector from the COM of Mcm2-7 to the H3/H4 tetramer to calculate the elevation (φ) and azimuthal (θ) angles (Fig. 3H). When the H3/H4 tetramer did not associate with Cdc45, the vector tended to orient to the direction with an elevation angle of $43° \pm 20°$ and an azimuthal angle of $-11° \pm 48°$ (Fig. 3I, J). On the other hand, when the H3/H4 tetramer was associated with Cdc45, the vector tended to orient to the direction with an elevation angle of $25° \pm 13°$ and an azimuthal angle of $25° \pm 12°$ (Fig. 3K, L). Therefore, the elevation angle shifted from $43° \pm 20°$ to $25° \pm 13°$ upon Cdc45 association while the azimuthal angle shifted from $-11° \pm 48°$ to $25° \pm 12°$. The region where the DNA strands (Fig. 3E, G) or the H3/H4 tetramers (Fig. 3J, L) frequently resided was surrounded by a polygon surface. In the simulations with a charged Cdc45 acidic loop (WT), the region where the H3/H4 tetramers resided (Fig. 3K, L) has an about five times larger overlap with the region of the leading strand (Fig. 3F, G, 0.70 %) than that of the lagging strand (Fig. 3D, E, 0.14 %). Thus, the increase in co-orientation probability contributes to the leading strand bias of histone recycling in the Cdc45-mediated pathway.

## The number of RPA on the lagging strand regulates the strand bias

Previous studies have shown that the number of RPA on the lagging strand inversely correlates with the concentration of Pol α[46,55]. Although a single-molecule fluorescence imaging study revealed that the number of RPA is $1.5 \pm 0.3$ at 70 nM Pol α[46], it may fluctuate in a cellular environment. To investigate the dependency of the number of RPA on recycling, we performed molecular dynamics simulations for $1 \times 10^8$ steps using the new sets of initial structures: one without RPA (0-RPA, 16 nt ssDNA gap on lagging strand) and one with a single molecule of RPA (1-RPA, 36 nt ssDNA gap) (Fig. 4A, B). As a result, the

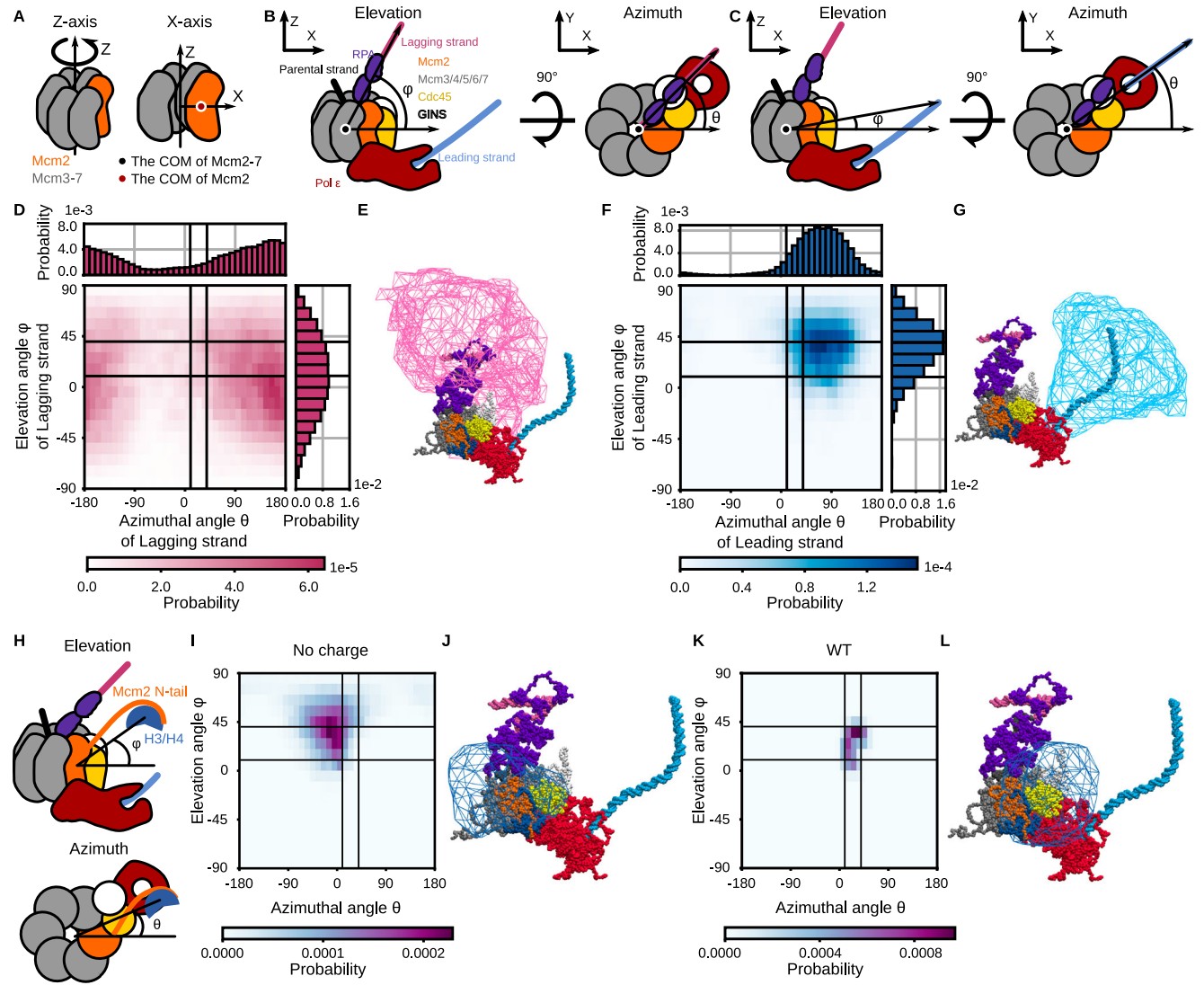

**Fig. 3 | Distributions of orientations of the replicated strands and the H3/H4 tetramer in the simulations of the replicated-DNA-engaged replisome. Source data are provided as a Source Data file. A** Definition of X- and Z-axes. **B, C** Schematic illustrations of the replicated-DNA-engaged replisome, which explains the definition of the elevation ($\varphi$) and azimuthal ($\theta$) angles to describe the lagging (**B**) and leading (**C**) strand orientation. The color scheme is the same as in Fig. 1B. **D, F** 1D and 2D probability distributions of the lagging (**D**) and leading (**F**) strand orientation calculated from the simulation trajectories until recycling. **E, G** Iso-surfaces of the spatial probability density of the lagging (**E**) and leading (**G**) strand orientation. The iso-value was set to 0.0001 [Å⁻³]. **H** Schematic illustrations of the replicated-DNA-engaged replisome explaining the definition of the elevation ($\varphi$) and azimuthal ($\theta$) angles to describe the H3/H4 tetramer orientation. **I, K** 2D distributions of the H3/H4 tetramer orientation calculated from trajectories of the simulations in the absence (**I**) and presence (**K**) of charges in the Cdc45 acidic loop. **J, L** Iso-surfaces of spatial probability distributions of the H3/H4 tetramer orientation in the simulation in the absence (**J**) and presence (**L**) of charges in the Cdc45 acidic loop. The iso-value was set to 0.0001 [Å⁻³]. **D, F, I, K** The black auxiliary lines, which encircle the populated area in (**K**), mark the azimuthal angles of 10° and 40° and the elevation angles of 10° and 40°.

Mcm2 N-tail deposited the H3/H4 tetramer on the leading or lagging strand via the Cdc45-mediated or unmediated pathway (Fig. 4C, Supplementary Fig. 6A–D) regardless of the number of RPA.

In the 0-RPA case, 56% (28/50) of the trajectories showed recycling to the leading or lagging strands, while 2% (1/50) and 42% (21/50) resulted in binding to the parental strand and no binding, respectively. Of the 28 recycling trajectories, 54% (15/28) and 46% (13/28) showed recycling via the Cdc45-mediated and unmediated pathways, respectively. 100% (13/13) of the recycling trajectories via the Cdc45-unmediated pathway resulted in the deposition of the H3/H4 tetramer on the lagging strand, while 73% (11/15) of the trajectories via the Cdc45-mediated pathway ended up with the deposition on the leading strand.

In the 1-RPA case, 38% (38/100) of the trajectories showed recycling to the leading or lagging strands, while 23% (23/100) and 39% (39/

100) resulted in binding to the parental strand and no binding, respectively. Of the 38 recycling trajectories, 61% (23/38) and 39% (15/38) showed recycling via the Cdc45-mediated and unmediated pathways, respectively. 80% (12/15) of the trajectories of recycling via the Cdc45-unmediated pathway resulted in the deposition of the H3/H4 tetramer on the lagging strand, while 65% (15/23) of the trajectories via the Cdc45-mediated pathway ended up with the deposition on the leading strand.

These statistics showed that the lagging-strand recycling bias was mitigated as the number of RPA associated with the lagging strand increased (Fig. 4D). To get further insights, we classified the trajectories based on the pathways. Note that the H3/H4 tetramer was deposited mainly on leading and lagging strands via the Cdc45-mediated and unmediated pathways, respectively, as in the 2-RPA case. The analysis showed that the leading strand bias via the

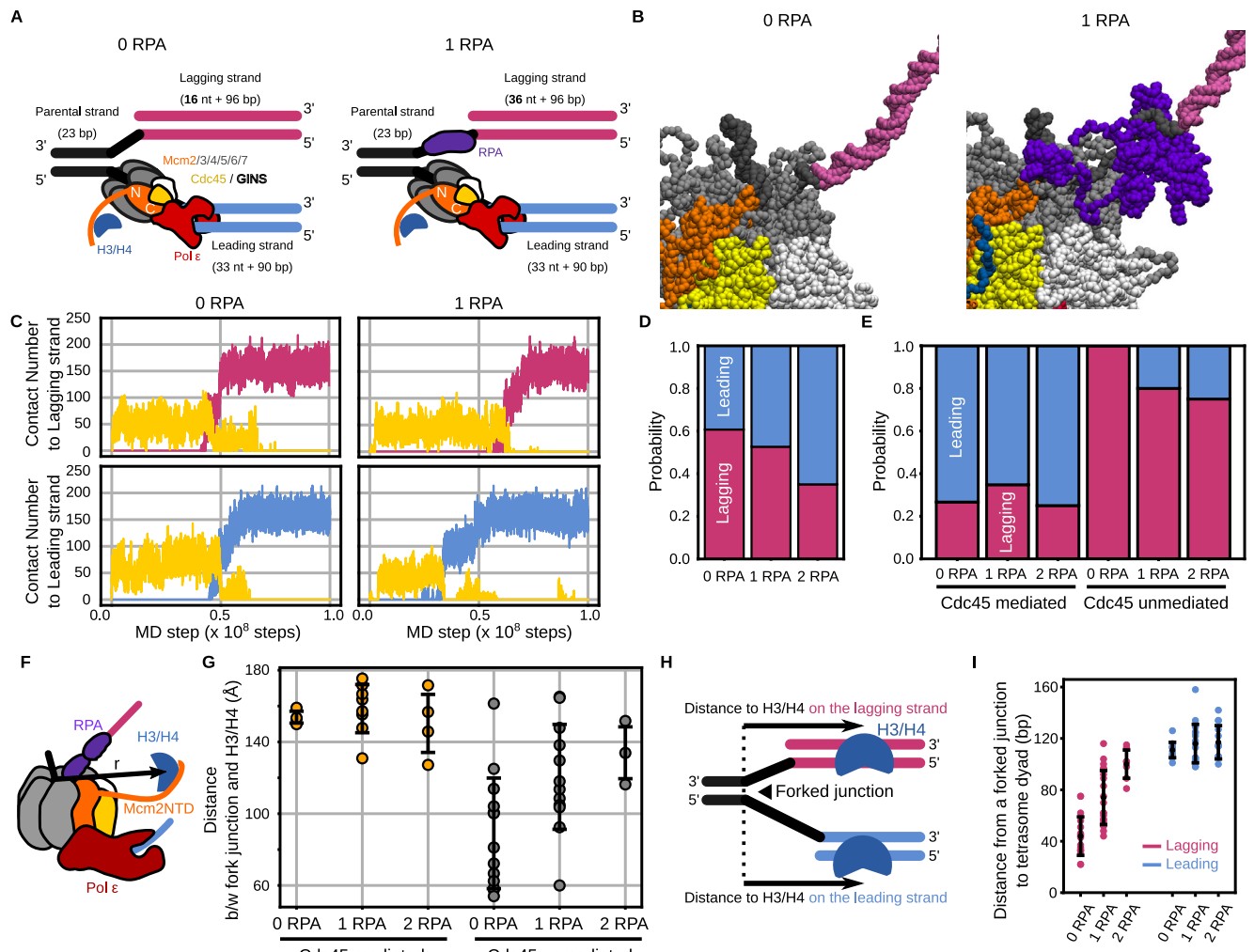

**Fig. 4 | The H3/H4 tetramer was deposited on the replicated strands by the Mcm2 N-tail in the simulations of the replicated-DNA-engaged replisome with zero or one RPA.** Source data are provided as a Source Data file. **A** Schematic illustration of the initial structures. The color scheme is the same as in Fig. 1B. **B** Initial structures with zero (left) and one (right) RPA molecule. **C** The number of residues in the H3/H4 tetramer contacting the lagging strand (pink), the leading strand (cyan), and Cdc45 (yellow) in the case of zero (left) and one (right) RPA molecule. **D** Ratios of the replicated strands on which the H3/H4 tetramer was deposited in the simulations with zero, one, and two RPA molecules. **E** The same as in (**D**), but trajectories were classified into the Cdc45-mediated and unmediated pathways. **F** Schematic illustration explaining the definition of the 3D distance from

the fork junction to the H3/H4 tetramer. **G** Distances from the fork junction to the H3/H4 tetramer at the moment of recycling to the lagging strand via the Cdc45-mediated pathway (orange; $n = 4$, 8, and 4) and the Cdc45-unmediated pathway (gray; $n = 13$, 12, and 3) in the simulations with zero, one, and two RPA molecules. The error bars represent the mean ± standard deviation. **H** Schematic illustration explaining the definition of the distance from the fork junction to the destination location. **I** Distances from the fork junction to the position of the tetrasome dyad on the lagging (pink; $n = 17$, 20, and 7) and leading strand (cyan; $n = 11$, 18, and 13) in the simulations with zero, one, and two RPA molecules. The error bars represent the mean ± standard deviation.

Cdc45-mediated pathway was unaffected, while the lagging strand bias via the Cdc45-unmediated pathway was mitigated (Fig. 4E). Together, the simulations suggested that RPA association with a lagging strand promotes the leading strand bias by inhibiting recycling to the lagging strand via the Cdc45-unmediated pathway. Previous in vitro and in vivo studies have shown that the ssDNA length on the lagging strand varies between 0 to 2000 nt depending on the concentration of Pol α-primase complex[46,55], which is required to initiate the lagging strand synthesis. This knowledge and our simulation results collectively suggested that the lagging strand synthesis initiation rate modulates the strand bias. Notably, a recent ChIP-NChAP study has shown that the parental histones preferentially bind to the strand replicated first[56] in line with this suggestion.

To confirm whether RPA binding affects the dynamics of the H3/H4 tetramer, we analyzed the orientation of the H3/H4 tetramer until recycling in the 0- and 1-RPA case as in the 2-RPA case above. The H3/H4 tetramer which did not bind to Cdc45 oriented

with the azimuthal angle of −13° ± 44°, −18° ± 50°, and −18° ± 34° and with the elevation angle of 35° ± 23°, 39° ± 22°, and 39° ± 19° in 0-, 1-, and 2-RPA case (Supplementary Fig. 7A–C). On the other hand, the H3/H4 tetramer which bound to Cdc45 oriented with the azimuthal angle of 20° ± 11°, 7° ± 12°, and 25° ± 13° and with the elevation angle of 9° ± 13°, 19° ± 13°, and 25° ± 12° in the 0-, 1-, and 2-RPA case (Supplementary Fig. 7D–F). We further calculated the three-dimensional distance between the H3/H4 tetramer and the fork junction until recycling (Fig. 4F). The distances were 133 ± 32 Å, 129 ± 35 Å, and 137 ± 25 Å in the 0-, 1-, and 2-RPA case, respectively, when the H3/H4 tetramer dissociated from Cdc45 (Supplementary Fig. 7G–I). On the other hand, the distances were 150 ± 14 Å, 150 ± 15 Å, and 150 ± 15 Å, whereas the H3/H4 tetramer associates to Cdc45. Therefore, these analyzes indicated that RPA binding did not significantly affect the region where the H3/H4 tetramer migrates until recycling. We performed a similar analysis on the dynamics of lagging strands and found that the direction

of lagging strands is also unaffected by RPA binding (Supplementary Fig. 8A, B & Fig. 3D).

To investigate the reason for the bias mitigation, we analyzed the distance between the fork junction and the H3/H4 tetramer at the moment of recycling to the lagging strand. (Fig. 4F). The distance was $89 \pm 31$ Å, $121 \pm 29$ Å, and $134 \pm 14$ Å via the Cdc45-unmediated pathway, while $154 \pm 3$ Å, $159 \pm 13$ Å, and $150 \pm 16$ Å in the 0-, 1-, and 2-RPA case (Fig. 4G). In other words, in the absence of RPA, the distance was widely distributed from 60 Å to 160 Å, whereas as the number of RPA molecules increased, the average value increased, and the range became narrower, which may be caused by occlusion and extension of ssDNA regions by RPA (Supplementary Fig. 7J, K). These results suggested that the migration area of the histones that can be recycled is constrained as the number of RPA molecules increases. It is reasonable to think that this constraint is one reason RPA prevents recycling to the lagging strand via the Cdc45 unmediated pathway.

Next, we focused on the destination location where the H3/H4 tetramer was recycled. We defined the distance from the fork junction as the number of nucleotides from the fork junction to the dyad of a recycled tetrasome (Fig. 4H). As the number of RPA associated with the lagging strand increased from zero to two, the distance increased from $43 \pm 15$ to $100 \pm 11$ nt for recycling to the lagging strand (Fig. 4I). On the other hand, the distance was unaltered in a significant way for recycling to the leading strand. This is as expected since the binding of RPA to the lagging strand shifts the naked dsDNA region required for recycling upstream (Figs. 1B, 4A). Remarkably, simulation results revealed that the H3/H4 tetramer could be recycled to the lagging strand even in the presence of two RPA molecules, equal to or slightly more than the average number ($1.5 \pm 0.3$) in physiological conditions[46,47].

### Pol ε association affects the destination location for the leading strand recycling

In the replicated-DNA-engaged replisome structure[27,28], Pol ε attaches to the CMG helicase complex and bends the leading strand (Figs. 1D, 5A). To investigate the role of DNA bending in recycling, we performed 50 runs of simulations of the replicated-DNA-engaged replisome without Pol ε and RPA molecules for $1 \times 10^8$ steps (Fig. 5A). As expected, Pol ε restrained the leading strand orientation in the simulations (Fig. 5B, C), and the leading strand oriented to the direction with an elevation angle of $34° \pm 25°$ and an azimuthal angle of $71° \pm 49°$ in the presence of Pol ε (CMGE) while with an elevation angle of $-24° \pm 33°$ and an azimuthal angle of $-178° \pm 95°$ in its absence (CMG).

In the simulations without Pol ε, 74% (37/50) of the trajectories showed recycling to the leading or lagging strands, while 6% (3/50) and 20% (10/50) resulted in binding to the parental strand and no binding, respectively. Of the 37 recycling trajectories, 38% (14/37) and 62% (23/37) showed recycling via the Cdc45-mediated and the Cdc45-unmediated pathways, respectively. The H3/H4 tetramer preferred to be recycled via the Cdc45-unmediated pathway in the absence of Pol ε, while it preferred to be recycled via the Cdc45-mediated pathway in the presence of Pol ε. 79% (11/14) of the trajectories of recycling via the Cdc45-mediated pathway resulted in the deposition of the H3/H4 tetramer on the leading strand, while 91% (21/23) of the trajectories via the Cdc45-unmediated pathway ended up with the deposition on the lagging strand (Fig. 5D). These statistics are comparable to those from the simulations in the presence of Pol ε (69% and 100%, respectively, Fig. 5D). Together, DNA bending by Pol ε did not significantly affect the strand bias.

Next, we performed the analysis using the vector from the COM of Mcm2-7 to the H3/H4 tetramer to calculate the elevation (φ) and azimuthal (θ) angles to investigate the effect of Pol ε on the movement of the H3/H4 tetramer. The vector tended to orient to the direction with an elevation angle of $15° \pm 19°$ and an azimuthal angle of $15° \pm 26°$ in the presence of Pol ε (Fig. 5E). On the other hand, the vector tended to orient to the direction with an elevation angle of $17° \pm 27°$ and an azimuthal angle of $12° \pm 32°$ in the absence of Pol ε (Fig. 5F). Therefore, Pol ε caused a minuscule change in the orientation of the H3/H4 tetramer.

To investigate the destination location in the absence of Pol ε, we calculated the distance from the fork junction to the recycled position as defined above (Fig. 4H). The distances were $78 \pm 13$ and $40 \pm 11$ nt for the leading and lagging strands, respectively (Fig. 5G). Thus, the distance for the leading strand in the absence of Pol ε decreased compared to its presence while not significantly altered for the lagging strand. Since we did not change the leading ssDNA length regardless of Pol ε existence, and the Pol ε only occludes the ssDNA region, its occlusion alone cannot explain the alteration of the destination location. Instead, the Pol ε binding extended the ssDNA region and moved the ssDNA-dsDNA junction away from the H3/H4 tetramer in the Cdc45-mediated pathway, where the tetramer is preferably recycled to the leading strand (Fig. 5H, I). In fact, we measured the distance between the junction and Cdc45, which turned out to be $202 \pm 18$ Å and $129 \pm 26$ Å in the presence and the absence of Pol ε. This result indicated that the temporal dissociation of Pol ε, e.g., upon polymerase exchange[46], may alter the destination location and, hence, the gene regulation in daughter cells. In our structural model, the length of the ssDNA gap on the leading strand was 33 nt and passed through the central channel of Mcm2-7 and the catalytic subunit of Pol ε. A recent study has shown that this ssDNA region is elongated due to uncoupling between DNA unwinding and leading strand synthesis upon replication stress[45]. This ssDNA looping between Mcm2-7 and Pol ε may make the destination location further away from the fork junction. This consideration leads to an attractive hypothesis that the replication speed modulates the transcriptional programs in daughter cells via altered nucleosome positioning.

## Discussion

In this study, we performed molecular dynamics simulations of a yeast DNA replication machinery containing Mcm2-7, Cdc45, GINS, Pol ε, and RPA, bound to an H3/H4 tetramer and replicated DNA to visualize the structural trajectory from the H3/H4 tetramer bound to Mcm2 until recycled to the replicated strands. The simulations and the in vitro replication assays combinatorically showed that Mcm2 can directly deposit the H3/H4 tetramer onto the replicated strands without additional factors such as histone chaperones. Interestingly, the simulations also suggested that the H3/H4 tetramer is more likely recycled to the leading and lagging strands in the Cdc45-mediated and unmediated pathways, respectively. Also, RPA binding to the lagging strand inhibited recycling via the Cdc45-unmediated pathway, tilting the strand bias toward the leading strand. On the other hand, Pol ε binding to the leading strand did not significantly alter the strand bias but did affect the destination location on the leading strand.

The most prominent prediction from the simulations is the Cdc45-mediated pathway of recycling. In the simulations, the H3/H4 tetramer was associated with the acidic loop of Cdc45 and was mainly recycled to the leading strand. The Native-PAGE assays supported that Cdc45 and an H3/H4 tetramer interact electrostatically. Future electron microscopy structures of the intermediate of histone recycling may prove that Cdc45 can capture a histone hexamer (or tetramer). The interaction may also have a regulatory role in the strand bias. Remarkably, the T189 and T195 residues in the acidic loop of Cdc45 are known to be phosphorylated, a prerequisite for recruiting Rad53, an S-phase checkpoint kinase[53]. These post-translational modifications lead to more negative charges in the acidic loop, which may enhance interactions with the H3/H4 tetramer. Also, the binding of Rad53 to the acidic loop may weaken the binding of the H3/H4 tetramer. The regulations of the strand bias of histone recycling by post-translational modifications and protein binding are a new paradigm of epigenetic inheritance, and experimental verification is strongly desired.

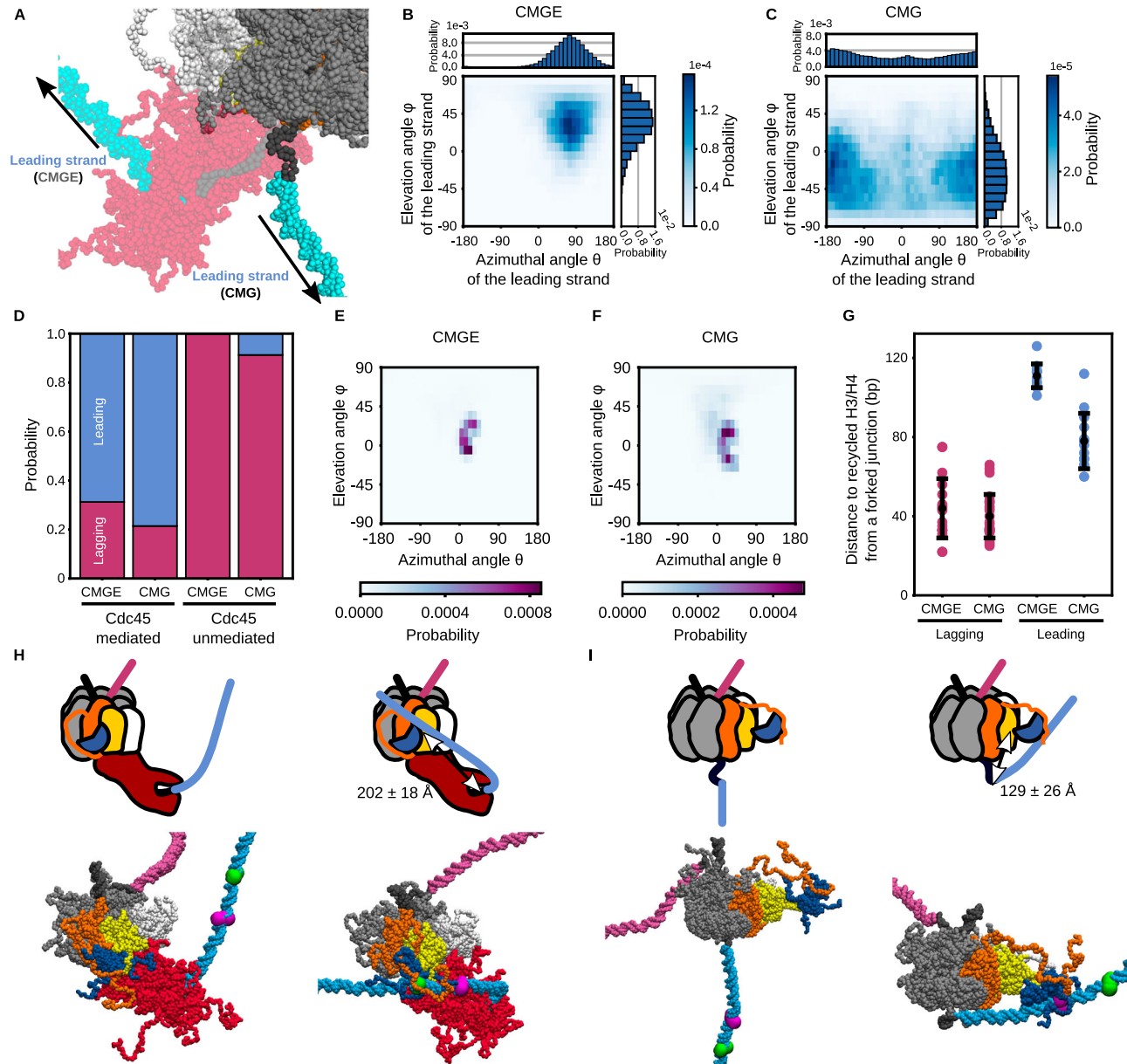

**Fig. 5 | The H3/H4 tetramer was deposited on the daughter strands by the Mcm2 N-tail in the simulations of the replicated-DNA-engaged replisome without Pol ε and RPA.** Source data are provided as a Source Data file. **A** The structure of CMG (opaque) superimposed to that of CMGE (CMG + Pol ε) (transparent). **B, C** 1D and 2D probability distributions of the leading strand orientation calculated from the simulations of CMGE and CMG. **D** Ratios of the replicated strands on which the H3/H4 tetramer was deposited in the simulations of CMGE and CMG. **E, F** 2D distributions of the H3/H4 tetramer orientation calculated from trajectories of the simulations of CMGE and CMG (**G**) Distances from the fork

junction to the deposited positions on the lagging (pink; $n = 17$ and 24) and leading strand (cyan; $n = 11$ and 13) in the simulations of CMGE and CMG. The error bars represent the mean ± standard deviation. **H, I** Schematic illustrations and representative structures of CMGE (**H**) and CMG (**I**). The initial structure and the structure at the exact moment of the H3/H4 deposition are shown on the left and right, respectively. The color scheme is the same as in Fig. 1B, D. Additionally, the green and magenta spheres on the leading strand represent the average recycled position in the CMGE and CMG simulations, respectively.

In recycling, either of the daughter strands took the H3/H4 tetramer away from the Mcm2 N-tail. Thus, the extent to which the strands probe the H3/H4 tetramer around a replication fork is critical. A recent study has proposed a theoretical model that diffusion of parental histones or DNA segments accounts for dispersed histone inheritance in active chromatin domains[57]. In this diffusion-driven model, histones can be recycled on any physically proximal DNA segment. Notably, our simulations are in line with the possibility that DNA segments far away from the fork junction associate with H3/H4 tetramer bound to the Mcm2 N-tail. Also, a previous in vitro single-molecule imaging suggested that histones can be recycled on the

destination location further than the DNA persistent length via DNA loop formation[58]. Therefore, the pathways visualized in the current simulations may contribute to the dispersion of epigenetic marks in active chromatin domains.

The previous studies using deep sequencing techniques such as SCAR-seq[14] and eSPAN[22] demonstrated that Mcm2 N-tail contributes to recycling, especially on the lagging strand, in the cellular condition. Consistent with these findings, our simulations reproduced the direct handover of the H3/H4 tetramer from the Mcm2 N-tail to the lagging strand. However, the current study also showed that the Mcm2 N-tail deposits the H3/H4 tetramer predominantly to the leading strand via

the Cdc45-mediated pathway. This result indicates that the factors missing in the current simulation setup regulate the predominant pathway in cellular conditions. Interestingly, recently solved electron microscopy structure of the intermediate state of histone recycling supported that a histone hexamer is captured by FACT bound to the N-terminal side of Mcm2-7[48], indicating that FACT may bias the predominant pathway to the lagging strand. Future studies should address the mechanisms of strand bias determination.

As a limitation of this study, the simulation system lacks the replisome components such as Ctf4, Pol α, Fen1, Lig1, and PCNA, the histone chaperones such as CAF-1, Asf1, FACT, and HJURP, and the chromatin remodelers such as INO80 and ISW1, which were suggested to cooperate with Mcm2 for histone recycling[18,19,22,26,48,59–61]. It is tempting to assume that these additional factors are decisive in choosing the dominant pathway. Also, whether the binding of the H3/H4 tetramer is limited only to the parental one, and if so, what is the molecular mechanism to accomplish it remains intriguing open questions. Notably, previous single-molecule imaging using *Xenopus laevis* egg extracts revealed that the recycling efficiency of the parental histones depends on the concentration of the newly synthesized histones[20], supporting that the pathways simulated in this study are used to deposit both parental and newly synthesized histones. Furthermore, the treatment of the inter-molecular interactions in our coarse-grained model is simple, which may underestimate the moderately strong interaction between the H3/H4 tetramer and RPA, which was experimentally detected, for example[62]. On this occasion, additional structural information may help improve the accuracy of the simulation results. Otherwise, the potential energy functions for hydrogen bonding and hydrophobic interactions can be incorporated[38,63]. Also, the simulations cannot account for the coupling among DNA unwinding, DNA synthesis, and histone recycling. Future studies can address this limitation by incorporating the potential switching procedure to model protein conformational change upon ATP hydrolysis[64]. Despite these limitations, the simulations provided insights and experimentally testable predictions on the molecular mechanism, which is vital for elucidating the intracellular histone recycling mechanisms regulated by the cooperation of various histone chaperones. The most direct procedure to test this prediction is to reconstitute histone recycling in vitro[19] and measure the recycling frequency and strand bias. It is highly hoped that such a method will be established.

## Methods
### Coarse-grained molecular dynamics simulations
For proteins (Cdc45, Mcm2-7, GINS, Pol ε, RPA, and an H3/H4 tetramer), we used the AICG2+ model[40] representing one amino acid as one bead located at $C_\alpha$ atom position. The following paragraphs describe how we modeled the initial structures of the CMG helicase complex (Cdc45 + Mcm2-7 + GINS), DNA polymerase ε (Pol ε), Replication protein A (RPA), and the H3/H4 tetramer.

To prepare the initial structure of the CMG helicase complex, we used the cryo-EM structure (PDB ID: 6U0M)[27] as a reference. The CMG helicase comprises three proteins: Cdc45, Mcm2-7, and GINS[65] (Fig. 1A). The reference structure contains the residues 1–650 of Cdc45. Mcm2-7 is a protein consisting of six subunits: Mcm2/3/4/5/6/7. The reference structure contains the residues 1–868 of Mcm2, 17–738 of Mcm3, 177–838 of Mcm4, 24–693 of Mcm5, 103–840 of Mcm6, and 1–729 of Mcm7. GINS is a protein consisting of four subunits: Sld5, Psf1, Psf2, and Psf3. The reference structure contains the residues 3–294 of Sld5, 1–208 of Psf1, 3–200 of Psf2, and 3–193 of Psf3. We treated the residues 166–217 and 437–457 of Cdc45, 1–200 and 707–736 of Mcm2, 58–90, 142–150, 332–337 and 571–650 of Mcm3, 213–220, 470–497, 731–740, 780–792 and 839–850 of Mcm4, 104–129, 212–234, 306–318, 340–345 and 644–646 of Mcm5, 246–259, 415–427, and 484–509 of Mcm6, 32–58, 159–188, 217–219 and 387–392 of Mcm7, 3–53, 111–120

and 239–247 of Sld5, 33–49 of Psf2, and 30–32, 59–67 and 142–161 of Psf3 as intrinsically disordered regions. The initial conformations of the intrinsically disordered regions (IDRs) were generated using MODELER[66]. All IDRs in other protein models were similarly generated.

For Pol ε, we used the cryo-EM structure (PDB ID: 6HV9[28]) as a reference. Pol ε comprises four proteins: Pol2, Dpb2, Dpb3, and Dpb4[67] (Fig. 1A). The reference structure contains the residues 1308–2222 of Pol2 and 1–90 of Dpb2 interacting with the CMG helicase complex. To obtain the full-length Pol ε reference structure containing the residues 1–2222 of Pol2, 1–689 of Dpb2, 1–201 of Dpb3, and 1–196 of Dpb4, we superimposed the cryo-EM structure of Pol ε holoenzyme (PDB ID: 6WJV[67]) and Dpb2 (PDB ID: 6HV9[28]) to the reference structure above (PDB ID: 6U0M[27]). We treated the residues 1–30, 91–107, 215–233, 664–677, 1187–1269, 1393–1403, 1748–1783, 1977–1993, 2033–2042, 2073–2099, and 2122–2127 of Pol2, 91–169, 195–206, 234–265, 368–377 and 557–597 of Dpb2, 1–8 and 94–201 of Dpb3, 1–17 and 125–196 of Dpb4 as IDRs.

For the H3/H4 tetramer, we used the crystal structure of a nucleosome core particle (PDB ID: 1KX5[68]) as the reference structure. The amino acid sequence was taken from *Xenopus laevis* for parameter calibration purposes. However, because of the high degree of sequence similarity in histones (97.1 % for H3 and 99.0 % for H4), the impact of the sequence on conclusions obtained from the coarse-grained simulations should be minimal. The model was the same as that previously employed to study the nucleosome dynamics upon a collision with a DNA translocase in the presence[43] and absence[69] of a histone chaperone. We treated the residues 1–32 of H3 and 1–23 of H4 as IDRs.

For RPA, the partial crystal, cryo-EM, and homology-modeled structures were connected by flexible linker regions with MODELER[66]. RPA is a protein consisting of three subunits: Rfa1, Rfa2, and Rfa3 (Fig. 1A). The reference structure of Rfa1 comprises the partial structures of DBD-F (residues 1–132, PDB ID: 5OMB[70]), DBD-A (residues 181–294, PDB ID: 1YNX[71]), DBD-B (residues 295–419, PDB ID: 1JMC[72], homology model), and DBD-C (residues 442–619, PDB ID: 6I52[73]). The reference structure of Rfa2 comprises the partial structures of the winged helix domain (residues 205–273, PDB ID: 4OU0[74], homology model) and DBD-D (residues 32–182, PDB ID: 6I52[73]). The reference structure of Rfa3 comprises the structure of DBD-E (residues 1–122, PDB ID: 6I52[73]). The structures of Rfa1, Rfa2, and Rfa3 were assembled by superimposing DBD-C of Rfa1, DBD-D of Rfa2, and DBD-E of Rfa3 to the heterotrimeric cryo-EM structure (PDB ID: 6I52[73]). We treated the residues 133–180 and 420–441 of Rfa1, 1–31, and 183–204 of Rfa2 as IDRs.

For DNA, we used the 3SPN.2 model[41], in which one nucleotide was represented as three beads located at the centroid of base, sugar, and phosphate groups. The replicated-DNA structure was modeled by superimposing the ideal B-form DNA structure generated using 3DNA[75] to the forked DNA in the cryo-EM structure (PDB ID: 6U0M[27]), which comprises 23 bp, 90 bp, and 96 bp of dsDNA for the parental, leading, and lagging strands and 33 nt and varying length (16 nt in 0-RPA, 36 nt in 1-RPA, and 56 nt in 2-RPA) of ssDNA for the leading and lagging strands. The length of the ssDNA gap on the leading strand was 33 nt and passed through the central channel of Mcm2-7 and the catalytic subunit of Pol ε. Pol ε fixes the orientation of the leading strand dsDNA. Thus, the gap size appears to have little, if any, effect on strand bias in histone recycling. The structure-based potential was applied to stabilize the B-form DNA structure and to reproduce the persistence length of ds and ssDNA, the melting temperature, and the hybridization rate.

The potentials for the excluded volume and electrostatic interactions were applied to the inter-molecular interactions. On top of them, the structure-based potential was applied to the protein residue pairs or the residue-nucleotide pairs which form contact in the experimentally solved structures to stabilize the cryo-EM structure of the RPA-ssDNA complex (PDB ID: 6I52[73]), the crystal structure of the

Pol ε-DNA complex (PDB ID: 4M8O[76]), and the crystal structure of the H3/H4 dimer-Mcm2 complex (PDB ID: 4UUZ[15]). The potential for the hydrogen bonding interactions was also applied to the interactions between the H3/H4 tetramer and DNA. The parameters of this potential were calibrated in the previous studies[38,39] to stabilize the canonical nucleosome structure and to reproduce the salt-concentration-dependent DNA unwrapping from a nucleosome.

The potential derived from Debye-Hückel's theory was applied to the electrostatic interactions. We arranged the partial charges on protein surface beads by the RESPAC algorithm[37] so that the model reproduced the electrostatic potential around the all-atom structures except for the beads in the disordered regions where we set +1$e$ charge on lysine and arginine residues, −1$e$ charge on aspartic acid and glutamic acid residues, and zero charge on the other residues. The charges of DNA phosphate groups were set to −0.6$e$ for intra-DNA interactions to model the counter ion condensation around the phosphate groups within the framework of the Debye-Hückel model. On the other hand, the phosphate charges were set to −1.0$e$ for protein-DNA interactions to account for releases of counter ions upon the association between DNA and a protein.

We performed Langevin dynamics simulations to integrate the equations of motion with a timestep of 0.3 CafeMol time units (-14.7 fs). Temperature and the friction constant were set to 300 K and 0.843, respectively. The monovalent ion concentration of the Debye-Hückel model was set to 300 mM, and the dielectric constant was set to 78.0. All the simulations were conducted using CafeMol3.2[77] (https://www.cafemol.org). It took ~2 weeks to compute the entire system for $1 \times 10^8$ steps using two CPU cores (Intel® Xeon® Gold 6326) in parallel. All output coordinates from the simulations were visualized using VMD1.9.3[78] (https://www.ks.uiuc.edu) or PyMol2.4[79] (https://www.python.org).

### Optimization of the potential for the interactions between Mcm2 and the H3/H4 tetramer

In addition to the potentials for the excluded volume and electrostatic interactions, the AICG2+ potential was applied for the inter-protein interactions between the Mcm2 histone-binding region (residues 69–121) and the H3/H4 tetramer based on the crystal structure (PDB ID: 4UUZ[15]). The non-local term in the AICG2+ potential function is given as

$$V_{native} = \sigma \sum_{i,j} \epsilon_{ij} \left[ 5 \left( \frac{r_{ij0}}{r_{ij}} \right)^{12} - 6 \left( \frac{r_{ij0}}{r_{ij}} \right)^{10} \right] \quad (1)$$

where $\sigma$ is a global scaling parameter, $\varepsilon_{ij}$ is a site-specific parameter between the $i$-th residue in Mcm2 and the $j$-th residue in an H3/H4 dimer, and $r_{ij0}$ and $r_{ij}$ are distances between the $i$-th and the $j$-th residue in the native and the simulated structure, respectively. Since this interaction plays a decisive role in recycling, we sought to carefully calibrate the scaling parameter $\sigma$ so that the experimentally measured dissociation constant is reproduced. Thus, we performed replica-exchange molecular dynamics simulations of the Mcm2 N-tail (residues 1–200) and the H3/H4 dimer in a sphere with a radius of 100Å for $1 \times 10^9$ steps with $\sigma$ set to 0.65, 0.70, and 0.75 (Supplementary Fig. 1A, B). The temperature of each of the ten replicas was set from 300 K to 390 K with a 10 K linear increment. To define the bound and unbound states, we calculated the $Q$-score of each simulation snapshot (Supplementary Fig. 1C). The $Q$-score was defined as the ratio of the number of residue-residue contacts to the number of natively formed contacts[15]. We considered that a residue pair forms a contact when these beads are within 1.2 times the distance in the native structure. Then, we defined the bound state as the state with $Q > 0$ because even partial formation of native contacts can imply binding in the experimental setup. Even when a cutoff of 10 Å was used for the minimum distance between Mcm2 and the H3/H4 dimer,

the binding free energy was −10.1 ± 0.3 kcal/mol, suggesting that our simulation results are robust to the cutoff definition. By comparing the binding free energy calculated from each simulation ($\Delta G^{\sigma=0.65} = -5.3 \pm 0.1$ kcal/mol, $\Delta G^{\sigma=0.70} = -10.0 \pm 0.3$ kcal/mol, and $\Delta G^{\sigma=0.75} = -15.7 \pm 0.5$ kcal/mol) and experimental measurement ($\Delta G^{expt} = -10.4$ kcal/mol), we decided to set $\sigma$ to 0.70, which best reproduced the experiment. The previous experiment suggested a binding free energy between the H3/H4 tetramer and dsDNA of around −12 kcal/mol[80], in line with the direct handover of the H3/H4 tetramer from the Mcm2 N-tail to the replicated strands.

### Experimental material preparations

The ARS1-W601 DNA used for in vitro DNA replication was chemically synthesized and cloned into a plasmid pEX-A2J2 vector. The ARS1-W601 linear DNA fragments (1155 bp) were obtained by Polymerase Chain Reaction (PCR). The budding yeast histone octamers (H3, H4, H2A, and H2B) and replication proteins (ORC, Cdc6, Cdt1-Mcm2-7, DDK, S-CDK, Sld2, Sld3-Sld7, Dpb11, Cdc45, GINS, PCNA, RFC, Mcm10, RPA, Polymerase α (Pol α), Polymerase ε (Pol ε), Polymerase δ (Pol δ), Ctf4, Csm3-Tof1 and Mrc1) were purified as described previously[51]. Cdc45-7HIS, Cdc45-7HIS, (e) and histone H3/H4 complex used for native gel electrophoresis assays (Supplementary Fig. 3A) was expressed and purified from *Escherichia coli* as also described previously[52,53,81].

### Replication assays

The nucleosomes were initially assembled on the ARS1-W601 linear DNA (1155 bp) by salt gradient dialysis as described previously[52]. In brief, 27.7 nM DNA and 354 nM histone octamer were mixed and dialyzed against 20 mM Tris-HCl (pH 7.5), 2 mM EDTA, 2 mM DTT, and 0.08% NP40 with a linear NaCl gradient from 2 M to 0.2 M. This was further dialyzed against 25 mM HEPES-KOH (pH 7.5), 0.5 mM EDTA, 1 mM DTT, 0.04% NP40. The nucleosome substrate was then diluted 2-fold in Mcm buffer [25 mM HEPES-KOH (pH 7.5), 1 mM dithiothreitol (DTT), 5 mM ATP, 7.5 mM Mg(OAc)$_2$, 5% glycerol, 0.01% (w/v) NP-40, 0.1 mg/ml BSA] containing 100 mM KOAc, 15 nM Orc, 60 nM Cdt1-Mcm2-7, and 30 nM Cdc6. After a 15-minute incubation at 30 °C, 50 nM DDK was added, and incubation was continued for 15 min. The reaction was then 4-fold diluted in replication buffer [25 mM HEPES-KOH (pH 7.5), 1 mM DTT, 5 mM ATP, 7.5 mM Mg(OAc)$_2$, 5% glycerol, 0.01% (w/v) NP-40, 0.1 mg/ml BSA, 0.1 mM CTP, 0.1 mM GTP, 0.1 mM TTP, 40 μM dATP, 40 μM dCTP, 40 μM dGTP, 25 μM dTTP and 15 μM biotin-dUTP] containing 60 mM KOAc, and proteins in final concentration of Orc (3.75 nM), Cdt1-Mcm2-7 (15 nM), Cdc6 (7.5 nM), DDK (10 nM), S-CDK (5 nM), Sld2 (30 nM), Sld3-Sld7 (20 nM), Dpb11 (30 nM), Cdc45 (40 nM), GINS (12.5 nM), Tof1-Csm3 (20 nM), Mrc1 (10 nM), Ctf4 (20 nM), RFC (20 nM), PCNA (20 nM), RPA (100 nM), Pol α (5 nM), Pol ε (2 nM), Pol δ (10 nM), and Mcm10 (5 nM). The reaction mixture (100 μl) was incubated at 30 °C for 20 min. Note that the final KOAc concentration was -100 mM. The reaction was adjusted to 50 mM NaCl and 2 mM CaCl$_2$, then further digested by MNase (1 U/ul) at 30 °C for 30 min. The reaction was mixed with 4 μL of 0.5 M EDTA, 2 μL of 10% (w/v) SDS, and 1 μL of 20 mg/mL protease K, then incubated at 37 °C for 20 min. The sample was mixed with 1/6 volume of native DNA dye [15% (w/v) ficol, 10 mM HEPES-KOH (pH 7.5), 0.05% orange G] and applied to 7.5 polyacrylamide gel electrophoresis in EzRun TG buffer (ATTO) at room temperature for 65 min at 21 mA. After dipping in 1/2x TBE for 5 min, the DNAs were transferred to the Zeta-Probe membrane (Bio-Rad) using a wet-transfer blotter at 80 V for 50 min at 4 °C. The membrane was crosslinked using a UV illuminator and soaked in blocking solution (Cytiva) at room temperature for 20 min. The membrane was incubated with Dylight680-conjugated streptavidin in TBS containing 0.1% (w/v) Tween 20 and 0.01% (w/v) SDS for 30 min. After washing the membrane three times with TBS+Tween+SDS, gel images were captured by the ChemiDoc Touch imager (Bio-Rad).

## Native polyacrylamide gel electrophoresis assays

We mixed 0.5 μM Cdc45-7HIS and 0–2 μM H3/H4 tetramer in 20 μL reaction buffer [25 mM HEPES-KOH (pH7.5), 1 mM EDTA, 10% Glycerol, and 0.05% Tween 20] with 150, 300, and 750 mM KCl for low-, medium-, and high-salt conditions, respectively. The reactions were incubated for 15 min at 30 °C and divided into two 10 μL aliquots. Each aliquot was run on 5–20% Tris-Glycine or sodium dodecyl sulfate (SDS)-polyacrylamide gels (ePAGEL HR; 2331970, ATTO) for 75 min at 21 mA/mV. Gels were stained with Coomassie Brilliant Blue (EzStain Aqua; 2332370, ATTO), were imaged using the iBright FL 1500 Imaging System, and were analyzed using Image J software[82].

## Reporting summary

Further information on research design is available in the Nature Portfolio Reporting Summary linked to this article.

## Data availability

The data that support the findings of this study are available within the article and Supplementary Information files. Source data are provided with this paper. The input and trajectory files have been submitted to the Biological Structure Model Archive (BSM-Arc) under BSM-ID BSM00050 (https://bsma.pdbj.org/entry/50). Source data are provided with this paper.

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

## Acknowledgements

We would like to thank Prof. Shoji Takada and the laboratory members of the theoretical biophysics laboratory at Kyoto University for discussions and assistance throughout this work. This work was supported by the Grant-in-Aid for Transformative Research Areas (24H00883; to T.T.), the grant from the Kyoto University Foundation (to T.T.), the grant from the Takeda Science Foundation (to T.T.), the grant from the Shimazu Science Foundation (to T.T.), the grant from the Inamori Foundation (to T.T.), the Grant-in-Aid for Japan Society for the Promotion of Science Fellows (22J21003; to F.N.), and the grant from the Ginpuu Foundation (to F.N. and T.T.).

## Author contributions

F.N., Y.M., and T.T. designed the project. F.N. performed the molecular dynamics simulations, the in vitro binding assay of Cdc45 and H3/H4 tetramer, and their analyses. Y.M. performed the in vitro DNA replication assays and their analyses. F.N., Y.M., and T.T. co-wrote the manuscript.

## Competing interests

The authors have no conflict of interest, financial or otherwise.
