## [Peer Review File · Nature Communications]

Molecular mechanism of parental H3/H4 recycling at a replication forkREVIEWER COMMENTS

Reviewer #1 (Remarks to the Author):

Summary:

This research conducted molecular dynamics (MD) simulations of a yeast replication machinery, including Mcm2-7, Cdc45, GINS, Pol ϵ , and H3/H4 tetramer, along with replicated DNA. It was found that the H3/H4 tetramer initially bound to Mcm2 is transferred onto the replicated strands, a process termed recycling. The study demonstrates that the H3/H4 tetramer can be deposited on either the leading or lagging strand via a pathway mediated by Cdc45 or one not involving Cdc45. Specifically, in the Cdc45-mediated pathway, the H3/H4 tetramer tends to be deposited on the leading strand. Furthermore, the presence of RPA (Replication Protein A) restricts the migration area of histone recycling, although recycling of histones was observed even in the presence of two RPA molecules. Additionally, it was observed that Pol ϵ orients the leading strand in a suboptimal direction for recycling purposes. The research question appears intriguing as it seeks to comprehend the molecular mechanism underlying the recycling of the H3/H4 tetramer during replication. However, I have few comments and suggestions.

1. The study performed coarse-grained simulations of a very large system. This is very timely to push the boundaries of molecular simulations to address complex biological problems. While the study provides molecular mechanism its linkage with experimental studies should be strengthened beyond providing testable hypotheses.

2. In their model, they considered only electrostatic interactions and excluded volume interactions. Despite experimental examination of the binding affinity between cdc45 and the H3/H4 tetramer, they noted that the stable association could be attributed to hydrophobic interactions. The model could have been enhanced and made less dependable by incorporating these hydrophobic interactions.

3. What is the initial structure of the replication machinery? Is it based on experimental observations? For example, in the initial conformation, the H3/H4 tetramer was already bound to Mcm-2, does the initial conformation bias the observations obtained in the study, can authors comment on that? For example, they comment that in 100% of the simulation trajectories they obtained H3/H4 tetramer bound to Mcm2 associated with cdc45 at least once. Is not it based on the initial conformation?

4. In 20% of the simulation runs, Mcm2 placed H3/H4 on the replicated strands, while in the rest, it put H3/H4 on the parental strand. However, the main focus of the paper is on depositing H3/H4 on the replicated strand. Additionally, they noted that in 3 out of 100 simulation runs, H3/H4 didn't bind to cdc45, suggesting a pathway not involving cdc45. In both cases, I wonder how statistically significant these observations are. Could the lack of interaction in those three simulations be due to random chance?

5. The remarks about changes in azimuthal and elevation angles supporting this bias might not be easily understood by all readers. Could the schematic in Figure 3, which explains these angles, be simplified further? Even in the text, the discussion about these angles could be made easier to understand, in my opinion.

6. The section titled "Modelling of replicated-DNA-engaged replisome binding to H3/H4 tetramer" in the results (page 3, lines 83-158) provides detailed methods and would be better suited for inclusion in the methods section. Additionally, I think the discussion could be improved and condensed.

7. The manuscript should better cite studies of coarse-grained model of protein-DNA systems.

Reviewer #2 (Remarks to the Author):

This study presents a novel understanding of the molecular mechanism of histone recycling at a replication fork. The simulation study reveals two distinct recycling pathways: Cdc45-mediated and unmediated pathways. Moreover, the study uncovers the regulatory role of RPA binding to the

ssDNA in the ratio recycled to each strand. The proposed mechanism awaits experimental validation. The simulation study, employing a set of finely calibrated force fields, demonstrates the H3/H4 tetramer transition to the parental, leading, or lagging strand. This transition will occur when the binding free energy of the H3/H4 tetramer and MCM2 is designed to be higher than that of H3/H4 and dsDNA. Thus, the transition is coordinated in advance. However, the study also reveals unexpected, insightful events. The simulations show that the Cdc45 acidic loop, previously invisible in cryo-EM structural analysis due to its flexibility, plays a crucial role in histone recycling and regulates the transition ratio to the lagging or leading strand. The study shows that the spatial distribution of H3/H4 tetramers changes depending on the interaction with cdc45, determining to which strand the tetramer moves. In addition, the impact of RPA and Pol eta is discussed.

Before accepting the paper, several points should be clarified.

The authors discussed the length of the ssDNA gap on the leading strand, which can take a variety of lengths. The authors should clarify why 33 nt was used in their molecular configuration (Figure 1). The length will affect the ratio of the H3/H4 tetramer transition to the leading or lagging strand.

Line 111. Supplementary Figure 1 does not have the binding free energy of the Mcm2 and H3/H4 tetramer. It seems Figure 2I. The binding free energy calculated from the coarse-grained simulation highly depends on the definition. The authors need to clarify why the bound state is the state with $Q > 0$. In addition, the figure denotes the H3/H4 dimer-Mcm2 interaction, not the tetramer. In addition, it is helpful to understand the simulation results if the binding free energy of H3/H4 tetramer and dsDNA obtained by the simulation is provided.

Line 135. The authors used 1KX5 as the reference structure in which the histones are from *Xenopus laevis*. It is not clear if the histones used in the simulation are from budding yeast as used in their experiment.

Figure 2I legend does not match the figure.

Line 166-171. The current estimate of the H3/H4 tetramer association time to the dsDNA, derived from a fitting, is considered unreliable. The authors must re-evaluate the fitting error to ensure a more accurate estimation of the validity of the study's conclusions. This is particularly important as the coarse-grained simulation may overestimate the real recycling time. If this is the case, the authors are encouraged to rationalize why the time is estimated.

L 263. No definition for the coordinate system is provided. The x-axis is defined as the vector from the center of mass of Mcm2-7 to that of Mcm2, but how is the y-axis or z-axis defined?

Line 263. The DNA persistent length is typically around 500 bp but is described as 150 bp. Is this length specific to this coarse-grained simulation?

Line 385. Figure 5 shows the distance between the junction and the H3/H4 tetramer, but the text says the distance between the junction and Cdc45. It seems that the figure is wrong.

Lines 389-408 and Fig.6

This paragraph does not strengthen the authors' argument and can be excluded from the paper, though I guess the authors might start to study this molecular configuration.

In the figure legends, the authors should explain what the error bars denote wherever they appear (e.g., Figs. 2I and 4G, I).

Responses to Reviewer #1

Comment 1:

This research conducted molecular dynamics (MD) simulations of a yeast replication machinery, including Mcm2-7, Cdc45, GINS, Pol ϵ , and H3/H4 tetramer, along with replicated DNA. It was found that the H3/H4 tetramer initially bound to Mcm2 is transferred onto the replicated strands, a process termed recycling. The study demonstrates that the H3/H4 tetramer can be deposited on either the leading or lagging strand via a pathway mediated by Cdc45 or one not involving Cdc45. Specifically, in the Cdc45-mediated pathway, the H3/H4 tetramer tends to be deposited on the leading strand. Furthermore, the presence of RPA (Replication Protein A) restricts the migration area of histone recycling, although recycling of histones was observed even in the presence of two RPA molecules. Additionally, it was observed that Pol ϵ orients the leading strand in a suboptimal direction for recycling purposes. The research question appears intriguing as it seeks to comprehend the molecular mechanism underlying the recycling of the H3/H4 tetramer during replication. However, I have few comments and suggestions.

Response 1:

We would like to thank Reviewer 1 for carefully reviewing this manuscript and for being interested in reading it. We revised the manuscript in response to the comments and the suggestions below. We firmly believe that the revised manuscript is of quality worth being published in *Nature Communications*.

Comment 2:

The study performed coarse-grained simulations of a very large system. This is very timely to push the boundaries of molecular simulations to address complex biological problems. While the study provides molecular mechanism its linkage with experimental studies should be strengthened beyond providing testable hypotheses.

Response 2:

The reviewer correctly pointed out the need to strengthen the linkage between our simulation findings and experimental studies. Recent experimental studies using deep sequencing techniques

such as SCAR-seq [Petryk et al. (2018) Science 361:1389] and eSPAN [Gan et al. (2018) Mol Cell 72:140] have demonstrated the involvement of Mcm2 N-tail in histone recycling to lagging strand. Consistent with these findings, our simulations reproduced the direct handover of H3/H4 from Mcm2 N-tail to the lagging strand. This point should have been emphasized in the manuscript. Therefore, we revised the sentences in the Discussion section to “The previous studies using deep sequencing techniques such as SCAR-seq¹⁴ and eSPAN²² demonstrated that Mcm2 N-tail contributes to recycling, especially on the lagging strand, in the cellular condition. Consistent with these findings, our simulations reproduced the direct handover of the H3/H4 tetramer from the Mcm2 N-tail to the lagging strand. However, the current study also showed that the Mcm2 N-tail deposits the H3/H4 tetramer predominantly to the leading strand via the Cdc45-mediated pathway. This result indicates that the factors missing in the current simulation setup regulate the predominant pathway in cellular conditions.” Furthermore, a new electron microscopy structure of the intermediate state of histone recycling supported that a histone hexamer is captured by FACT bound to the N-terminal side of Mcm2-7. Future electron microscopy structures may reveal that Cdc45 captures a histone hexamer (or tetramer) as the simulations suggested. To emphasize this point, we added the new sentence, “Interestingly, recently solved electron microscopy structure of the intermediate state of histone recycling supported that a histone hexamer is captured by FACT bound to the N-terminal side of Mcm2-7⁴⁸, indicating that FACT may bias the predominant pathway to the lagging strand. Future studies should address the mechanisms of strand bias determination.” and “Future electron microscopy structures of the intermediate of histone recycling may prove that Cdc45 can capture a histone hexamer (or tetramer).” to the Discussion section. The most powerful method to test the hypothesis obtained from the simulations in this study is to reconstitute histone recycling *in vitro* and measure its recycling frequency and strand bias. We added the sentence, “The most direct procedure to test this prediction is to reconstitute histone recycling *in vitro*¹⁹ and measure the recycling frequency and strand bias. It is highly hoped that such a method will be established.” to the Discussion section to emphasize the importance of establishing such procedures. The changes here will enrich the discussion and enhance the manuscript’s impact.

Comment 3:

In their model, they considered only electrostatic interactions and excluded volume interactions. Despite experimental examination of the binding affinity between cdc45 and the H3/H4 tetramer, they noted that the stable association could be attributed to hydrophobic interactions. The model

could have been enhanced and made less dependable by incorporating these hydrophobic interactions.

Response 3:

Incorporating hydrophobic interactions into the model would likely lead to a more accurate representation. However, it should be noted that while the HPS model [Dignon et al. (2018) *PLoS Comput. Biol.* 14(1): e1005941] is currently one of the most reliable coarse-grained models for representing hydrophobic interactions, it has limitations in representing hydrophobic interactions within folded regions of Cdc45. Additionally, as evident from the new supplementary figure (Supplementary Figure S4) added to the revised manuscript, the surface of Cdc45 is predominantly filled with hydrophilic residues. This suggests that the contribution of hydrophobic interactions in the interaction with histones is limited. We believe that these facts justify not incorporating hydrophobic interactions into our coarse-grained model at this stage. To clarify this point, we have modified the sentence in the revised manuscript: "However, as evident from Supplementary Figure S4, the surface of Cdc45 is predominantly filled with hydrophilic residues, supporting that the contribution of hydrophobic interaction is not dominant, if any, and justifying not incorporating the interactions into our coarse-grained model."

Comment 4:

What is the initial structure of the replication machinery? Is it based on experimental observations? For example, in the initial conformation, the H3/H4 tetramer was already bound to Mcm-2, does the initial conformation bias the observations obtained in the study, can authors comment on that? For example, they comment that in 100% of the simulation trajectories they obtained H3/H4 tetramer bound to Mcm2 associated with cdc45 at least once. Is not it based on the initial conformation?

Response 4:

We want to clarify that all initial structures used in our simulations are based on experimental data, specifically X-ray crystallography and cryo-electron microscopy structures, as thoroughly explained in the Method section. These experimental structures include the structure of the H3/H4 tetramer binding to the Mcm2 N-tail, which has been observed in both gel electrophoresis [Foltman et al. (2013) *Cell Reports* 3:892] and cryo-electron microscopy [Li et al. (2024) *Nature* 627:890]

studies. Notably, the cryo-electron microscopy structure we refer to depicted the architecture of endogenous replisomes. This suggests the relevance of interactions between the H3/H4 tetramer and the Mcm2 N-tail as intermediate structures in histone-recycling reactions within cells. Of course, other pathways are possible but they were not considered here. In the revised manuscript, we commented on this by adding the sentence, “**The recent cryo-EM structure of an endogenous replisome⁴⁸ suggested the relevance of interactions between the H3/H4 tetramer and the Mcm2 N-tail as intermediate structures in histone recycling reactions within cells. Here, we focused on the recycling pathways in which the H3/H4 tetramer binds to the Mcm2 N-tail, although other pathways may also be possible.**” to the Result section.

We acknowledge that the initial conformation may influence subsequent dynamics in molecular simulations. However, our simulations showed that H3/H4 underwent repeated association to and dissociation from Cdc45 (Supplementary Figure S2E & S6C), suggesting structural relaxation from the initial state occurs significantly faster than the timescale of histone recycling reactions. Furthermore, H3/H4 demonstrates extensive sampling across spatial domains in our simulations (Figure 3I-3L). Based on these points, our simulation results are robust with respect to the initial structures. To clarify this point, we added the sentence “**The repeated dissociation from and association with Cdc45 suggested that the simulation result is robust to the initial structure.**” to the Result section.

Comment 5:

In 20% of the simulation runs, Mcm2 placed H3/H4 on the replicated strands, while in the rest, it put H3/H4 on the parental strand. However, the main focus of the paper is on depositing H3/H4 on the replicated strand. Additionally, they noted that in 3 out of 100 simulation runs, H3/H4 didn't bind to cdc45, suggesting a pathway not involving cdc45. In both cases, I wonder how statistically significant these observations are. Could the lack of interaction in those three simulations be due to random chance?

Response 5:

As indicated in Supplementary Figure S2B, our simulation results, while constrained by current computational limits, suggested that extending simulation durations could increase the number of recycling trajectories. Moreover, we hypothesized that mechanisms within cells, such as

downstream nucleosomes or histone chaperones, likely accelerate recycling. Despite these limitations in our simulation length and setups, we obtained 20 recycling trajectories, enough to statistically analyze the pathways to the replicated strands. Regarding the simulations where H3/H4 did not bind to Cdc45, these instances might reflect random occurrences rather than a dominant pathway. Nevertheless, the simulation results suggested that the Cdc45 unmediated pathway is statistically rare but is physically feasible. To clarify these points, we added the sentences “Below, we focused on the 20 recycling trajectories to statistically analyze the molecular pathway of histone recycling.” and “The rare occurrence of the Cdc45-unmediated pathway suggested that the path is physically feasible but is statistically rare.” to the Result section.

Comment 6:

The remarks about changes in azimuthal and elevation angles supporting this bias might not be easily understood by all readers. Could the schematic in Figure 3, which explains these angles, be simplified further? Even in the text, the discussion about these angles could be made easier to understand, in my opinion.

Response 6:

We appreciate the suggestion to simplify the explanation of azimuthal and elevation angles in Figure 3 and the text. We have revised Figure 3A to include a more precise depiction of the Z-axis and the X-axis. Additionally, we have provided a more precise explanation of these angles in the revised manuscript, aiming to improve accessibility for all readers: “The Z-axis is aligned with the direction of the rotation axis of Mcm2-7, and the X-axis is perpendicular to the Z-axis, passing through the COM of Mcm2 (Figure 3A). The Y-axis is perpendicular to both X- and Z-axes. The angle between the vectors and the X-axis on the X-Z plane is defined as the elevation angle φ , and the angle between the vectors and the X-axis on the X-Y plane is defined as the azimuthal angle θ (Figure 3B &C).” We believe these changes have significantly enhanced the clarity of our discussion regarding azimuthal and elevation angles.

Comment 7:

The section titled "Modelling of replicated-DNA-engaged replisome binding to H3/H4 tetramer" in the results (page 3, lines 83-158) provides detailed methods and would be better suited for inclusion in the methods section. Additionally, I think the discussion could be improved and condensed.

Response 7:

We have relocated the detailed methods described in the Result section to the Methods section. Also, we have condensed the Discussion section from 9 to 5 paragraphs to improve clarity and conciseness. We would like to thank Reviewer 1 for the valuable feedback, which helped us enhance the quality of our manuscript.

Comment 8:

The manuscript should better cite studies of coarse-grained model of protein-DNA systems.

Response 8:

We agree on citing previous studies applying coarse-grained models for protein-DNA systems to highlight that our computational approach is intended to visualize the molecular trajectories of H3/H4 recycling. To clarify these points, we added the sentence with citations, “**Coarse-grained molecular dynamics simulations have successfully visualized the dynamics of protein-DNA complexes in long-time scale, providing insights into molecular mechanisms^{31–35}.**”

Responses to Reviewer #2

Comment 1:

This study presents a novel understanding of the molecular mechanism of histone recycling at a replication fork. The simulation study reveals two distinct recycling pathways: Cdc45-mediated and unmediated pathways. Moreover, the study uncovers the regulatory role of RPA binding to the ssDNA in the ratio recycled to each strand. The proposed mechanism awaits experimental validation. The simulation study, employing a set of finely calibrated force fields, demonstrates the H3/H4 tetramer transition to the parental, leading, or lagging strand. This transition will occur

when the binding free energy of the H3/H4 tetramer and MCM2 is designed to be higher than that of H3/H4 and dsDNA. Thus, the transition is coordinated in advance. However, the study also reveals unexpected, insightful events. The simulations show that the Cdc45 acidic loop, previously invisible in cryo-EM structural analysis due to its flexibility, plays a crucial role in histone recycling and regulates the transition ratio to the lagging or leading strand. The study shows that the spatial distribution of H3/H4 tetramers changes depending on the interaction with cdc45, determining to which strand the tetramer moves. In addition, the impact of RPA and Pol ϵ is discussed. Before accepting the paper, several points should be clarified.

Response 1:

We would like to thank Reviewer 2 for carefully reading and reviewing the manuscript. We revised the manuscript in responses to the comments below. We firmly believe that the revised manuscript is ready to be published in *Nature Communications*.

Comment 2:

The authors discussed the length of the ssDNA gap on the leading strand, which can take a variety of lengths. The authors should clarify why 33 nt was used in their molecular configuration (Figure 1). The length will affect the ratio of the H3/H4 tetramer transition to the leading or lagging strand.

Response 2:

We chose to use 33 nt in our study based on the minimum length required for the single-strand DNA region to traverse the Mcm2-7 channel and reach the active center of Pol ϵ . Pol ϵ fixes the orientation of the leading strand dsDNA. Thus the gap size appears to have little, if any, effect on strand bias in histone recycling. To clarify this point, we added the sentence, “**The length of the ssDNA gap on the leading strand was 33 nt and passed through the central channel of Mcm2-7 and the catalytic subunit of Pol ϵ . Pol ϵ fixes the orientation of the leading strand dsDNA. Thus, the gap size appears to have little, if any, effect on strand bias in histone recycling.**” to the Method section.

Comment 3:

Line 111. Supplementary Figure 1 does not have the binding free energy of the Mcm2 and H3/H4 tetramer. It seems Figure 2I. The binding free energy calculated from the coarse-grained simulation highly depends on the definition. The authors need to clarify why the bound state is the state with $Q > 0$. In addition, the figure denotes the H3/H4 dimer-Mcm2 interaction, not the tetramer. In addition, it is helpful to understand the simulation results if the binding free energy of H3/H4 tetramer and dsDNA obtained by the simulation is provided.

Response 3:

We have revised the text to specify that Supplementary Figure 1 does not depict the binding free energy but rather illustrates the results of our temperature replica exchange simulations of Mcm2 and the H3/H4 dimer (revised text: “Therefore, in this study, we performed temperature replica exchange simulations of Mcm2 and an H3/H4 dimer associating to and dissociating from each other with varying parameters (Supplementary Figure 1) and selected the one that reproduced the experimental binding free energy (Simulation: -10.00 ± 0.26 kcal/mol, Experiment: -10.45 ± 0.04 kcal/mol¹⁵).”).

Regarding the definition of the bound state with $Q > 0$, we utilized this criterion because even partial formation of native contacts can imply binding in the experimental setup. Even when a cutoff of 10 Å was used for the minimum distance between Mcm2 and the H3/H4 dimer, the binding free energy was -10.1 ± 0.3 kcal/mol, suggesting that our simulation results are robust to the cutoff definition. To clarify this point, we added the sentence, “Then, we defined the bound state as the state with $Q > 0$ because even partial formation of native contacts can imply binding in the experimental setup. Even when a cutoff of 10 Å was used for the minimum distance between Mcm2 and the H3/H4 dimer, the binding free energy was -10.1 ± 0.3 kcal/mol, suggesting that our simulation results are robust to the cutoff definition.” to the Methods section.

The main text incorrectly stated that this was a replica exchange simulation of the Mcm2 N-tail and H3/H4 tetramer, which has been corrected in the revised manuscript as above. In reality, Mcm2 N-tail and H3/H4 dimer interact with each other, so the simulation results were compared with the experimental binding free energy of Mcm2 N-tail and H3/H4 dimer. We would like to express our apologies to the reviewers for the confusion caused by this error.

As for the binding free energy between the H3/H4 tetramer and dsDNA, while direct calculation from simulations remains challenging due to multiple binding sites of the tetramer on dsDNA, we note that the previous experiment suggested binding energy around -12 kcal/mol [Andrews et al.

(2010) Mol Cell 37:834], consistent with our simulation findings that Mcm2 N-tail directly hand over the H3/H4 tetramer to the replicated strands. To clarify this point, we added the sentence, “The previous experiment suggested a binding free energy between the H3/H4 tetramer and dsDNA of around -12 kcal/mol⁷⁹, in line with the direct handover of the H3/H4 tetramer from the Mcm2 N-tail to the replicated strands.” to the Methods section.

Comment 4:

Line 135. The authors used 1KX5 as the reference structure in which the histones are from *Xenopus laevis*. It is not clear if the histones used in the simulation are from budding yeast as used in their experiment.

Response 4:

Reviewer 2 correctly pointed out that the histone sequence used in the simulation is from *Xenopus laevis*. Past studies used this sequence to calibrate parameters to match experimental results [Niina et al. (2017) *PLoS Comput. Biol.* 13(12): e1005880, Brandani et al. (2021) *PLoS Comput. Biol.* 17(7):e1009253]. It is not straightforward to perform a similar calibration for *Saccharomyces cerevisiae* histones, as several corresponding experiments have yet to be performed. However, the sequence of H3 and H4 in *Xenopus laevis* and *Saccharomyces cerevisiae* is highly conserved at 97.1 % and 99.0 % similarity, respectively, and it is thought that the effect of sequence changes on intermolecular interactions is small. In particular, the impact on conclusions obtained from the simulations with the coarse-grained resolution should be minimal. We clarified this point by adding the sentence, “The amino acid sequence was taken from *Xenopus laevis* for parameter calibration purposes. However, because of the high degree of sequence similarity in histones (97.1 % for H3 and 99.0 % for H4), the impact of the sequence on conclusions obtained from the coarse-grained simulations should be minimal.” to the Methods section.

Comment 5:

Figure 2I legend does not match the figure.

Response 5:

We fixed this error. We would like to express our apologies to the reviewers for the confusion caused by this error.

Comment 6:

Line 166-171. The current estimate of the H3/H4 tetramer association time to the dsDNA, derived from a fitting, is considered unreliable. The authors must re-evaluate the fitting error to ensure a more accurate estimation of the validity of the study's conclusions. This is particularly important as the coarse-grained simulation may overestimate the real recycling time. If this is the case, the authors are encouraged to rationalize why the time is estimated.

Response 6:

We have removed the statement "roughly corresponding to ~1.5 microseconds" to avoid implying a precise mapping to real-world time from coarse-grained molecular dynamics simulations. We acknowledge that accurately estimating such timescales is challenging, as Reviewer 2 stated in the comment.

Additionally, we have modified the sentence to read: “We estimated that it would take $> 1 \times 10^9$ steps for the H3/H4 tetramer to associate with the parental, leading, or lagging strand in all the trajectories (Supplementary Figure 2B).” This change emphasizes that our conclusions are not critically dependent on precise time scale estimation.

Comment 7:

L 263. No definition for the coordinate system is provided. The x-axis is defined as the vector from the center of mass of Mcm2-7 to that of Mcm2, but how is the y-axis or z-axis defined?

Response 7:

We would like to express our apology for this ambiguity. We added the sentence, “The Z-axis is aligned with the direction of the rotation axis of Mcm2-7, and the X-axis is perpendicular to the

Z-axis, passing through the COM of Mcm2 (Figure 3A). The Y-axis is perpendicular to both X- and Z-axes.” to the Result section.

Comment 8:

Line 263. The DNA persistent length is typically around 500 bp but is described as 150 bp. Is this length specific to this coarse-grained simulation?

Response 8:

Under typical conditions, the persistence length of dsDNA is 50 nm [e.g., Geggier et al. (2010) PNAS 107:15421], equivalent to approximately 150 base pairs.

Comment 9:

Line 385. Figure 5 shows the distance between the junction and the H3/H4 tetramer, but the text says the distance between the junction and Cdc45. It seems that the figure is wrong.

Response 9:

We fixed this error by changing the figure representation. We would like to express our apologies to the reviewers for the confusion caused by this error.

Comment 10:

Lines 389-408 and Fig.6. This paragraph does not strengthen the authors' argument and can be excluded from the paper, though I guess the authors might start to study this molecular configuration.

Response 10:

As Reviewer 2 correctly pointed out, we have removed the paragraph and Fig. 6 as they do not strengthen the main argument of the paper. We deeply appreciate the suggestion, and this molecular configuration could be explored in future studies in different contexts.

Comment 11:

In the figure legends, the authors should explain what the error bars denote wherever they appear (e.g., Figs. 2I and 4G, I).

Response 11:

We would like to express our apology for the lack of information. The error bars represent mean \pm standard deviation. The figure legends were revised accordingly.

REVIEWERS' COMMENTS

Reviewer #1 (Remarks to the Author):

All my comments are adequately addressed in the revised manuscript

Reviewer #2 (Remarks to the Author):

The authors have satisfactorily addressed all the concerns raised by the reviewers. However, the numbering for supplementary figures is not consistent. Some are referred to as S5, while others are referred to as 5. Please ensure consistent numbering throughout the document.

Responses to Reviewer #1

Comment 1:

All my comments are adequately addressed in the revised manuscript.

Response 1:

We would like to thank Reviewer 1 again for carefully reviewing this manuscript.

Responses to Reviewer #2

Comment 1:

The authors have satisfactorily addressed all the concerns raised by the reviewers. However, the numbering for supplementary figures is not consistent. Some are referred to as S5, while others are referred to as 5. Please ensure consistent numbering throughout the document.

Response 1:

We also would like to thank Reviewer 2 for carefully reviewing the manuscript. We unified the notation.